# ADAPTABLE SYMBOLIC MUSIC INFILLING WITH MIDI-RWKV

## ABSTRACT

Existing work in automatic music generation has mostly focused on end-to-end systems that generate either entire compositions or continuations of pieces, which are difficult for composers to iterate on. The area of computer-assisted composition, where generative models integrate into existing creative workflows, remains comparatively underexplored. In this study, we address the tasks of model style adaptation and multi-track, long-context, and controllable symbolic music infilling to enhance the process of computer-assisted composition. We present MIDI-RWKV, a small foundation model based on the RWKV-7 linear architecture, to enable efficient and coherent musical cocreation on edge devices. We also demonstrate that MIDI-RWKV admits an effective method of finetuning its initial state for style adaptation in the very-low-sample regime. We evaluate MIDI-RWKV and its state tuning on several quantitative and qualitative metrics with respect to existing models, and release model weights and code in the supplementary materials.

## 1 INTRODUCTION

Music is a millennia-old art form that provides a means of artistic and personal expression across cultures. Its composition requires substantial human effort, which has prompted work on foundation models for music generation (Ma et al., 2024). Many such works formulate music generation as a sequence modeling task, leveraging the autoregressive Transformer (Vaswani et al., 2017) to achieve remarkable performance. However, a gap remains between technical capabilities and practical needs.

Despite advancements in music generation, most systems remain difficult for composers to use effectively. Autoregressive symbolic music models, such as those by von Rütte et al. (2023); Yu et al. (2022); Lu et al. (2023), allow composers to easily manipulate their outputs, but are unable to regenerate parts of compositions. Symbolic music infilling models, like those by Ens & Pasquier (2020); Pasquier et al. (2025); Malandro (2023; 2024), allow selective regeneration, but have limited context windows that fail to model common long-range dependencies, and struggle with controllability and style capture (Tchemeube et al., 2023). Feedback from the 2020 AI Song Writing Contest (Huang et al., 2020) agrees that existing systems are insufficiently steerable and context-aware.

In this work, we propose MIDI-RWKV, a symbolic music infilling model designed with controllability and adaptability in mind. We employ the RWKV-7 architecture (Peng et al., 2025) to efficiently model long sequences and incorporate effective numerical attribute controls to condition generation. We train on the diverse GigaMIDI dataset (Lee et al., 2025) to create a well-rounded foundation model, and demonstrate that it admits a finetuning/style adaptation method, *state tuning*, that surpasses existing methods in the low-sample regime. We evaluate our model and finetuning method on objective metrics and a subjective listening test.

The main contributions of our work are as follows:

- We introduce MIDI-RWKV, a symbolic music infilling model based on the RWKV-7 architecture that enables controllable and easily adaptable computer-assisted music generation.
- We show our model compares favorably to competitors quantitatively and qualitatively.
- We demonstrate that our *state tuning* approach for MIDI-RWKV outperforms both the base model and LoRA fine-tuning in several low-sample style adaptation tasks.
- We provide the first empirically-supported analysis of state tuning dynamics (Appendix D.2).

Table 1: A comparison of recent multi-track symbolic music infilling systems. Context lengths differ in units and tokenization: superscript [1] uses REMI+, [2,3] are custom. *: Limited by system memory.

| Model | Fixed Schema | Maximum Sequence Length | Attribute Controls | Parameter Count |
|---|---|---|---|---|
| **MIDI-RWKV** | **no** | $\infty^{1,*}$ | **yes** | **35M** |
| MIDI-Mistral (Rizzotti, 2025) | no | 8192 tokens[1] | no | 42M |
| MIDI-GPT (Pasquier et al., 2025) | no | 2048 tokens[1] | yes | 20M |
| MMM (Ens & Pasquier, 2020) | no | 2048 tokens[1] | yes | 20M |
| Composer's Assistant (Malandro, 2023) | no | 3300 tokens[2] | yes | 54M |
| CA 2 (Malandro, 2024) | no | 3300 tokens[2] | yes | 192M |
| MusIAC (Guo et al., 2022) | yes | 16 bars[3] | yes | 30M |

## 2 RELATED WORK

**Symbolic music infilling.**    *Musical infilling* is the reconstruction of musical content from surrounding material. Musical continuation can be viewed as a special case of infilling without future context. Early approaches to infilling single tracks leveraged Markov chain Monte Carlo methods (Hadjeres et al., 2017), variational autoencoders (Pati et al., 2019), convolutional (Huang et al., 2018a) and recurrent (Roberts et al., 2019) neural networks, diffusion UNets (Min et al., 2023), and Transformers (Hsu & Chang, 2021). Recent multi-track models have by and large used Transformers (Table 1).

**Controllability.**    Useful generative models must be effectively controllable by the user (Newman et al., 2023). Symbolic music models broadly use numerical, categorical, or ordinal *attribute controls* to enable controllability. In Table 1, MMM provides controls on instrument choice, note density, and polyphony, MIDI-GPT conditions on those as well as style and note duration, Composer's Assistant (CA) offers polyphony controls in version 1 and eight controls in version 2 including note density, and MusIAC offers five controls including note density. MIDI-Mistral does not offer attribute controls. Continuation models such as FIGARO (von Rütte et al., 2023), Museformer (Yu et al., 2022), MuseCoco (Lu et al., 2023), and the Music Transformer (Huang et al., 2018b) also condition on various attribute controls, and Bhandari et al. (2025) condition directly on text input in a similar (though non-contrastive) vein to CLIP (Radford et al., 2021) for images.

**Style adaptation.**    Artists desire generative systems that can mimic their own style (Malandro, 2024; Bryan-Kinns et al., 2025). The most popular solution has been the use of small *inspiration sets*, in light of the limited examples available from an individual artist; Vigliensoni et al. (2022b) discuss the merits of this so-called *small data* approach. Studies of small data in the visual domain include Sobhan et al. (2024), Abuzuraiq & Pasquier (2024a), and Abuzuraiq & Pasquier (2024b).

In the music domain, Bryan-Kinns et al. (2025) highlight that practicing musicians tend toward the small data approach and offer suggestions for future model design. Vigliensoni et al. (2022a) present a system for live drum rhythm generation from small data. Sarmento et al. (2023) explore adaptation of a GuitarPro Transformer model to four individual guitar players. Malandro (2024) indicates the Composer's Assistant 2 (CA 2) infilling model admits low-data finetuning for individual users.

**Linear architectures.**    Traditional softmax attention (Vaswani et al., 2017) incurs $\mathcal{O}(n^2)$ time and space cost in the sequence length $n$, which becomes significant as sequence length grows. This has resulted in interest into the design of neural network architectures that afford $\mathcal{O}(n)$ complexity while still admitting parallelizability of training. Examples include state space models (Gu & Dao, 2023; Dao & Gu, 2024) and linear attention variants (Peng et al., 2023; Katharopoulos et al., 2020; Behrouz et al., 2024), many of which have rivaled or surpassed Transformer performance at equivalent model size and compute budget. These models, derived from recurrent neural networks (RNNs), can technically operate on infinitely long sequences—up to physical memory limits—whereas Transformers are typically restricted by the length of their trained positional embeddings. Many recent variations integrate the delta rule (Widrow & Hoff, 1960) into their state updates to train an expressive hidden state at test time (Schlag et al., 2021; Yang et al., 2024), including the RWKV-7 (Receptance-Weighted Key-Value 7) architecture (Peng et al., 2025) we use.

Figure 1: Comparison of one track of sheet music (above) with REMI+ (below). REMI tokens have black text, tokens unique to REMI+ have white text. TrackEnd is omitted because the track continues.

**State tuning.** The convention when modeling with RNNs is to set the initial hidden state to a zero vector. However, it has been known since the 1990s that this can be suboptimal, and that training the initial state as an additional variable can improve performance (Gers et al., 2002; Forcada & Carrasco, 1995). We refer to this as *state tuning*. The scant existing research on state tuning (Pitis, 2016; Wenke & Fleming, 2019) demonstrates useful changes in model dynamics; for instance, Mohajerin & Waslander (2017) show that a zero initial state corresponds to a steady-state system and that state tuning improves modeling of transient behavior. State tuning therefore presents promise for modern RNNs with expressive hidden states, such as those aforementioned, as their hidden states encode more information more expressively than those of early architectures (Peng et al., 2025).

## 3 MIDI-RWKV

In this section, we describe MIDI-RWKV and its components. Section 3.1 describes the encoding of symbolic music data into tokens. Then, we show how we address controllability in Section 3.2. Section 3.3 explains our infilling objective, and Section 3.4 explains state tuning for style adaptation.

### 3.1 DATA ENCODING

Because symbolic music has a natural discrete organization along the time dimension, it lends itself to modeling as a token sequence. We adopt the REMI+ encoding (von Rütte et al., 2023) to do so, an extension of REMI (Huang & Yang, 2020), as is standard for musical infilling (see Table 1). An example of a single track encoded using REMI+ is shown in Figure 1. REMI represents musical notes with tokens for bar breaks, position within a bar, tempo, pitch, and velocity, while REMI+ extends this to multiple tracks by adding tokens for track start and end, time signature, and program (instrument), where tracks are placed sequentially instead of interleaved.

Since one note takes several tokens in REMI+, we use byte-pair encoding (BPE) (Gage, 1994) to reduce the length of tokenized sequences and augment the vocabulary from 663 to 16000 tokens, motivated by subword tokenization in language modeling (Sennrich et al., 2016). We use the MidiTok library (Fradet et al., 2021) to perform efficient encoding/decoding and to train the BPE tokenizer.

### 3.2 ATTRIBUTE CONTROLS

We address controllability by formulating symbolic music infilling as a conditional sequence generation task, where the model must generate a sequence of musical tokens given surrounding context and control tokens. These control tokens each correspond to a musical attribute such as polyphony or note density, and are thus called *attribute controls*. The premise is that, given a numerical, categorical, or ordinal musical attribute $a$ for which we can compute a value from a musical excerpt $x$, we can teach a model the conditional relationship between $x$ and a token that encodes $a$ (Pasquier et al., 2025).

We use three attributes computed on a per-bar basis using MidiTok: *note density*, *note duration*, and *polyphony*. Note density represents the number of notes in a bar and ranges from 1 to 18, with an extra bin for 18+. Note duration represents which note types (whole, half, quarter, eighth, sixteenth) are present in a bar, using a binary token for each note type. Polyphony represents the minimum and maximum number of notes played simultaneously at any note onset time in the bar.

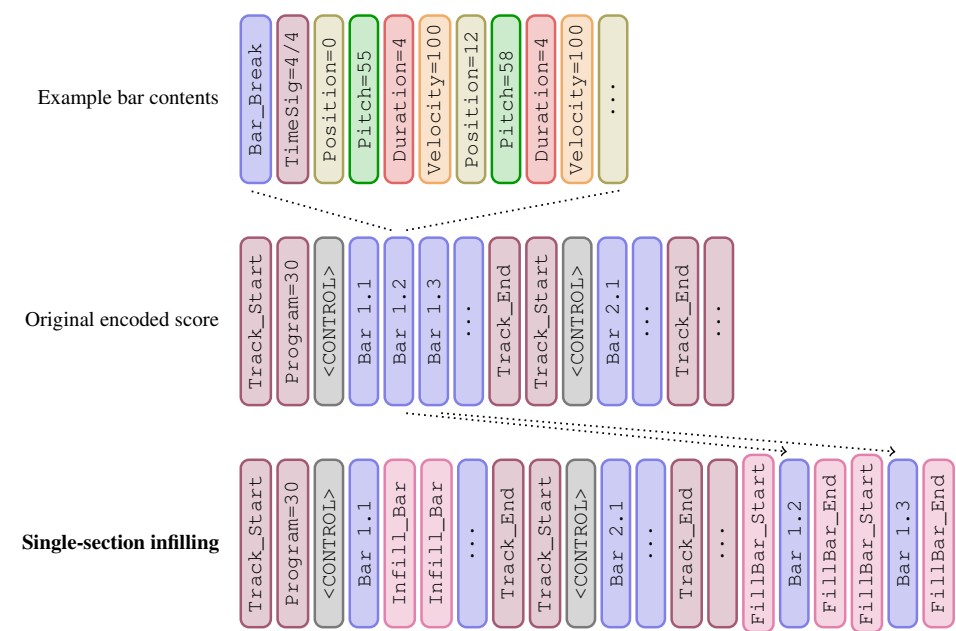

Figure 2: Our single-section infilling representation. A contiguous set of bars is masked and its content is moved to the end of the sequence. <CONTROL> represents a control token sequence.

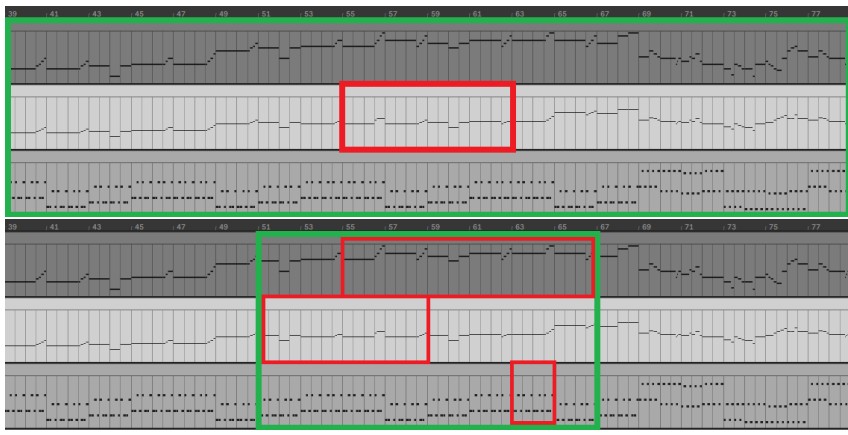

Figure 3: Example training samples for single-section infilling (above) and arbitrary masking pattern infilling (below). Measures to infill (masked) are outlined in red, the full context window in green.

These controls were designed in collaboration with Steinberg GmbH and evaluated in Cubase by Tchemeube et al. (2023), where they were found to be effective in practical usage. For monophonic instruments, higher density and lower duration tend to correspond to texture tracks, and vice versa to melodies; these were thus determined to be the most effective controls for controlling generation.

### 3.3 SINGLE-SECTION INFILLING

We adapt the Bar-Fill representation of MIDI-GPT (Pasquier et al., 2025) for our objective, as shown in Figure 2, due to its proven record in production deployment (Tchemeube et al., 2023). Bars to be infilled are masked by one `Infill_Bar` token per bar, and the content previously in those bars is moved to the end of the sequence and marked as *infill content*. Inference thus consists of providing the concatenated tracks with some masked bars, whereupon the model will generate the infill content. This allows the seq2seq objective, adopted by Malandro (2023; 2024) to great effect, to be emulated using a decoder-only Transformer, as RWKV-7 does not admit the notion of an encoder.

Our training sample format, which we deem *single-section infilling*, differs slightly from that of basic Bar-Fill. See Figures 2 and 3 for a visualization. We elect to infill only one contiguous section of masked bars per sample, although infilling in the bottom format in Figure 3—which we call an *arbitrary masking pattern*, due to allowing any combination of bars within the context window to be masked, and is the format in which real-world user requests would be received—can be achieved with multiple calls to the model, and is abstracted from the user by the inference pipeline. We explain the design decisions motivating single-section infilling in Appendix A. We denote the width of the infilling section as $N$ and the context window width as $C$; henceforth we use $C = 4N$ unless otherwise specified, chosen to best leverage RWKV-7's excellent long-context performance.

## 3.4 MODEL ADAPTATION WITH STATE TUNING

RWKV models, like other subquadratic architectures, maintain state vectors that encode information from previously processed tokens, which Transformers lack (the analog would be the KV cache). The initial state vectors are almost always set to zero, but can in theory be set to any vector or learned through a posttraining process. This presents an opportunity for style adaptation tasks, as we can train only the relatively small hidden states to condition on a particular style without extensive retraining. Specifically, given an *inspiration set* of $m$ training samples, we can freeze the model's parameters and optimize only the initial state vectors $\{\vec{h}_{0,i}\}_{i=1}^{L}$ for each of the $L$ layers using the cross-entropy loss. We explore this premise using the RWKV-PEFT library (Kang, 2025).

From the lens of dynamical systems, state tuning may be thought of as a biasing of the trajectory of the model through its state space by adjusting the initial conditions (Mohajerin & Waslander, 2017). Theoretically, this directs the hidden state evolution to operate within a new subspace of representation space that better corresponds to the finetuning data; in this sense, it extracts information the model has already learned and stored in its weights, rather than teaching it new information with weight updates. We provide empirical justification for this claim in Appendix D.2. We therefore expect state tuning to be most effective for adapting foundation models in domains with both global and local attributes, such as symbolic music, where fundamental music structures stay mostly constant across pieces but individual styles and techniques vary widely.

The most common approaches to finetuning are full-parameter finetuning and low-rank adaptation (Hu et al., 2022), so we find it pertinent to address the differences between those and state tuning. State tuning requires significantly fewer parameters compared to both—only $Ld$ parameters for a vector-valued hidden state, or $Ld^2$ for matrix-valued state, where $d$ is the hidden dimension—but does not allow for variability in trainable parameter count as LoRA does. Like LoRA adapters, multiple tuned state vectors can be used with the same model and shared at less expense than a full finetune, but unlike LoRA adapters, multiple cannot be used in conjunction.

## 4 EXPERIMENTAL SETUP

### 4.1 TRAINING

**Base model.** We train our base model using the RWKV-LM library (Peng, 2025) on the train set of the GigaMIDI dataset (Lee et al., 2025), comprising 1.05 million MIDI files. The model follows the "deep and narrow" paradigm suggested by Malandro (2023), using 12 RWKV-7 layers with head size 64, a hidden dimension of 384, and a feedforward dimension of 1,536, resulting in roughly 35 million trainable parameters. The weights are initialized as in Peng et al. (2025). We train for 48 epochs with the Adam optimizer and a cosine learning rate from 1e-4 to 1e-5, weight decay of 0.1, no dropout, and a batch size of 16, which takes 64 hours on $1\times$RTX 4090 with a sequence length of 4096.

**Finetuning experiments.** We perform finetuning using the RWKV-PEFT library (Kang, 2025) on 99 randomly selected songs from the POP909 dataset (Wang et al., 2020), leaving the remaining 810 for testing. We train and evaluate only on the melody track of each song, instead of training on all tracks, to more easily discern differences between finetuning strategies (being a more specialized task) and to facilitate a subjective listening test (melody being the easiest track to isolate aurally). We select this small training set to be a proxy for the small corpuses of composers who will likely be using our system. The scripts for both LoRA and state tuning are written in Triton (Tillet et al., 2019), allowing users on GPU-less edge devices to train in tractable time using the CPU backend for Triton.

We state tune with a high learning rate of 5e-2, no learning rate decay, and no dropout for 16 epochs, which trains 294 thousand parameters in about 4 minutes. We demonstrate that this high learning rate still results in stable training dynamics in Appendix D.1.

We also finetune comparisons using LoRA (Hu et al., 2022) for the same amount of time using a lower learning rate of 5e-4 and no dropout. Time is chosen as the unit of measure due to its importance in practical applications (Vigliensoni et al., 2022b; Bryan-Kinns et al., 2025). We train on the same train/test split as the first state tuned model. We train two adapters: one with $r = \alpha = 32$, training 2.7 million parameters in about 6 minutes, and one with $r = \alpha = 4$, training 331 thousand parameters in about 4.5 minutes. The latter is used for parameter parity with state tuning, and the former is more demonstrative of LoRA's theoretical performance. Because RWKV models' LoRA dynamics are known to be stable (Kang, 2025), we train only one of each. Experiments on other finetuning datasets, performed with the same procedure, are available in Appendix D.4.

While we aim to demonstrate that state tuning is superior to LoRA for MIDI-RWKV specifically, we also compare against LoRA-finetuned MIDI-GPT in Appendix D.3 and demonstrate that state-tuned MIDI-RWKV also surpasses MIDI-GPT with LoRA.

## 4.2 INFERENCE

We perform inference on all MIDI-RWKV generations with the `rwkv.cpp` library (RWKV, 2025) and sampling parameters of temperature=1.0, repetition penalty=1.2, top-k=20, and top-p=0.95, as suggested by Rizzotti (2025) and ablated in Appendix E.3. Generations with other models are performed using their recommended inference scripts and default sampling parameters.

Each base model was tested on a subset of 1000 songs of the GigaMIDI test set, on two objectives: our single-section infilling objective at $N = 2, 4, 8$ with $C = 4N$, and the random infilling objectives of Malandro (2023), where 50% of measures in a contiguous $N$-measure multi-track section are masked at random with $N = 8, 16$. Finetuned models were tested only on single-section infilling, with the same $N$ and $C$, on the 810 POP909 songs not in the finetuning train set.

Any prompts of lengths greater than the maximum input sequence length for Transformer comparison models were chunked for processing. Since MIDI-RWKV can operate on arbitrary sequence lengths during inference, we did not chunk its inputs; however, even with BPE, several examples in the $N = 8, C = 32$ task were over 8192 tokens in length and few were below 4096 tokens, which demonstrates the remarkable extrapolation abilities of RWKV-7 from its training sequence length.

## 4.3 METRICS

We evaluate on four standard objective metrics to evaluate MIDI-RWKV and comparable models using MIDI-Metrics (Rizzotti, 2025). More details may be found in Appendix C. All statistical tests are Wilcoxon signed rank tests with Holm-Bonferroni correction unless otherwise stated.

- **Content preservation (CP)** (Lu & Su, 2018) is the average cosine similarity between moving averages of pitch chroma vectors of corresponding bars of the original and infilled content, which measures style preservation of the original song. Higher is better.
- **Groove similarity (GS)** (Wu & Yang, 2020) is the average ratio of onset positions that match between corresponding bars of the original and infilled content, which measures preservation of rhythm. Higher is better.
- **Pitch class histogram entropy difference (PCHE)** (Wu & Yang, 2020) is the difference between the entropy of the pitch frequency vectors of corresponding bars of the original and infilled content, which measures tonality preservation. Lower is better.
- **F1 score (F1)** is the harmonic mean of precision and recall, measuring how well infilled notes match the original content. Higher is better.

We note, however, that ground truth-based sample-to-sample metrics are inherently limited measures of quality, being that music is fundamentally creative and subjective; a "good" infilling might diverge significantly from the original while maintaining musical coherence, such as an ABAB section being infilled to ABCB. We particularly do not trust F1 score, as the other three at least measure proxies for musicality instead of exact note matches, and only report it for consistency with other publications.

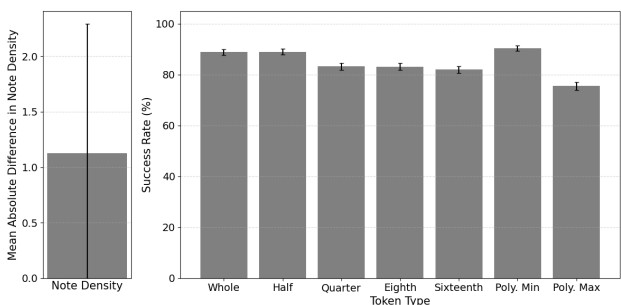

Figure 4: Evaluation of attribute control effectiveness. Left: Average absolute difference between real and intended note density. Right: Success rate of categorical control tokens.

Thus we augment the sample-to-sample metrics with a distribution-to-distribution metric, using StyleRank (Ens & Pasquier, 2019) to quantify stylistic similarity to the training data. Our experimental setup is identical to that of Pasquier et al. (2025), which we detail in Appendix B.3. Furthermore, we evaluate model adherence to provided attribute controls, which is simply calculated by computing the musical attributes of the generated content and comparing them against the input control tokens.

We also performed a subjective listening test with 28 participants to validate the effectiveness of state tuning, as the ground truth may not reflect the only reasonable infilling. For each of five prompts, participants ranked four anonymized clips (original, base model, LoRA $r = \alpha = 4$, and state-tuned) in order of overall preference. Further experimental details are available in Appendix B.4.

## 5 RESULTS

### 5.1 OBJECTIVE EVALUATION

**Base model comparison.** We compare our base model against three comparable models of Table 1: Composer's Assistant (CA), MIDI-GPT, and MIDI-Mistral. In particular, for DAW integration, we compare against models lightweight enough to run on composers' existing hardware, which frequently lack GPUs; we thus compare against other models in the 20-60M parameter range, with feasible CPU inference on edge devices. We thus defer comparison with CA2 to Appendix E.5, wherein we also compare to a diffusion baseline, Polyffusion (Min et al., 2023). We do not compare against MMM due to being superseded by MIDI-GPT or MusIAC due to input schema incompatibility.

Single-section infilling results are reported in Table 2 and random infilling in Table 3; entries are labeled "$N$-bar ModelName" and have "MIDI-" prefixes removed. MIDI-Mistral cannot operate on chunked inputs, so we stopped gathering results after single-section $N = 4$.

While the results are sometimes mixed, we conclude that MIDI-RWKV generally matches or outperforms comparable models on objective metrics in both single-section and random infilling, especially as task length grows. MIDI-RWKV soundly outperforms MIDI-GPT and MIDI-Mistral across all metrics, often by large margins, and compares favorably to Composer's Assistant, a 54% larger model. We also do not find F1 score a particularly compelling measure, as outlined in Section 4.3.

**Finetuning comparison.** We report finetuning results in Table 4, denoting by "LoRA #" a LoRA trained with $r = \alpha = \#$ and "state" a state tuned model. We conclude that state tuning on a small subset of the POP909 dataset generally outperforms the base model and LoRA on objective metrics.

**Attribute control effectiveness.** We evaluate adherence to attribute controls in Figure 4. The categorical control tokens are observed to be broadly effective, and the note density is on average around 1 note per bar of the desired amount. However, one must note that attribute control can go against consistency with context (e.g. rapidly swapping between high and low note densities); thus 100% success rate may not even be desirable, as the model must weigh adherence to the prompt against musical coherence. Evaluations on POP909 are available in Appendix E.4.

Table 2: Single-section infilling evaluation of models on the GigaMIDI test set.

| Model | CP ↑ | GS ↑ | PCHE ↓ | F1 ↑ |
|---|---|---|---|---|
| 2-bar RWKV | **0.606 ± 0.279** | 0.940 ± 0.062 | 0.477 ± 0.445 | **0.186 ± 0.176** |
| 2-bar CA | 0.420 ± 0.271 | **0.969 ± 0.050** | **0.364 ± 0.540** | 0.148 ± 0.306 |
| 2-bar GPT | 0.403 ± 0.300 | 0.964 ± 0.061 | 0.615 ± 0.404 | 0.072 ± 0.059 |
| 2-bar Mistral | 0.227 ± 0.216 | 0.905 ± 0.079 | 0.876 ± 0.688 | 0.026 ± 0.083 |
| 4-bar RWKV | **0.605 ± 0.243** | 0.939 ± 0.069 | **0.405 ± 0.399** | 0.122 ± 0.135 |
| 4-bar CA | 0.366 ± 0.258 | **0.960 ± 0.057** | 0.408 ± 0.684 | **0.233 ± 0.346** |
| 4-bar GPT | 0.374 ± 0.273 | 0.952 ± 0.024 | 0.508 ± 0.447 | 0.060 ± 0.091 |
| 4-bar Mistral | 0.011 ± 0.034 | 0.862 ± 0.095 | 0.800 ± 0.669 | 0.011 ± 0.034 |
| 8-bar RWKV | **0.596 ± 0.234** | **0.938 ± 0.072** | **0.317 ± 0.343** | 0.125 ± 0.156 |
| 8-bar CA | 0.410 ± 0.355 | 0.937 ± 0.074 | 0.492 ± 0.669 | **0.278 ± 0.402** |
| 8-bar GPT | 0.380 ± 0.260 | 0.950 ± 0.056 | 0.467 ± 0.372 | 0.064 ± 0.086 |

Table 3: Random infilling evaluation of models on the GigaMIDI test set.

| Model | CP ↑ | GS ↑ | PCHE ↓ | F1 ↑ |
|---|---|---|---|---|
| 8-bar RWKV | **0.694 ± 0.141** | **0.943 ± 0.022** | 0.300 ± 0.318 | 0.219 ± 0.153 |
| 8-bar CA | 0.474 ± 0.394 | 0.936 ± 0.087 | **0.281 ± 0.346** | **0.509 ± 0.421** |
| 8-bar GPT | 0.587 ± 0.262 | 0.934 ± 0.073 | 0.337 ± 0.319 | 0.062 ± 0.077 |
| 16-bar RWKV | **0.682 ± 0.164** | **0.940 ± 0.044** | **0.278 ± 0.268** | 0.342 ± 0.256 |
| 16-bar CA | 0.465 ± 0.366 | **0.940 ± 0.075** | 0.301 ± 0.388 | **0.578 ± 0.422** |
| 16-bar GPT | 0.576 ± 0.303 | 0.929 ± 0.066 | 0.332 ± 0.391 | 0.051 ± 0.081 |

Table 4: Single-section infilling evaluation of MIDI-RWKV finetunes on a POP909 test set. Different superscripts in the same column and section indicate statistically significant ($p < 0.05$) differences.

| | CP ↑ | GS ↑ | PCHE ↓ | F1 ↑ |
|---|---|---|---|---|
| 2-bar base | $0.316 \pm 0.182^a$ | $0.947 \pm 0.039^a$ | $0.497 \pm 0.415^a$ | $0.063 \pm 0.146^a$ |
| 2-bar LoRA 4 | $0.331 \pm 0.205^b$ | $0.949 \pm 0.040^a$ | $0.460 \pm 0.399^b$ | $\mathbf{0.074 \pm 0.103}^b$ |
| 2-bar LoRA 32 | $0.341 \pm 0.201^c$ | $0.944 \pm 0.043^b$ | $0.473 \pm 0.400^c$ | $0.070 \pm 0.129^b$ |
| 2-bar state | $\mathbf{0.351 \pm 0.269}^d$ | $\mathbf{0.952 \pm 0.037}^c$ | $\mathbf{0.439 \pm 0.407}^d$ | $0.073 \pm 0.190^b$ |
| 4-bar base | $0.293 \pm 0.159^a$ | $0.925 \pm 0.052^a$ | $0.433 \pm 0.348^a$ | $\mathbf{0.072 \pm 0.107}^a$ |
| 4-bar LoRA 4 | $0.338 \pm 0.152^b$ | $0.922 \pm 0.058^b$ | $0.368 \pm 0.299^b$ | $0.063 \pm 0.098^b$ |
| 4-bar LoRA 32 | $0.330 \pm 0.152^c$ | $0.918 \pm 0.060^c$ | $0.358 \pm 0.286^c$ | $0.065 \pm 0.102^b$ |
| 4-bar state | $\mathbf{0.345 \pm 0.152}^d$ | $\mathbf{0.928 \pm 0.043}^d$ | $\mathbf{0.355 \pm 0.312}^c$ | $0.070 \pm 0.105^a$ |
| 8-bar base | $0.287 \pm 0.116^a$ | $0.884 \pm 0.062^a$ | $0.314 \pm 0.276^a$ | $0.054 \pm 0.072^a$ |
| 8-bar LoRA 4 | $0.302 \pm 0.115^b$ | $0.885 \pm 0.057^a$ | $0.275 \pm 0.258^b$ | $\mathbf{0.056 \pm 0.069}^a$ |
| 8-bar LoRA 32 | $0.310 \pm 0.108^c$ | $0.883 \pm 0.058^a$ | $0.261 \pm 0.200^c$ | $0.053 \pm 0.080^a$ |
| 8-bar state | $\mathbf{0.352 \pm 0.168}^d$ | $\mathbf{0.908 \pm 0.065}^b$ | $\mathbf{0.244 \pm 0.339}^d$ | $0.054 \pm 0.107^a$ |

**StyleRank.** The StyleRank results are listed in Table 5, where "Sets Closer to Corpus than Corpus to Corpus" is as in Appendix B.3. Since $p > 0.05$ for all trials, we conclude that MIDI-RWKV is able to generate outputs consistent with the original material in style as quantified by StyleRank, even as objective length increases and the model has more room to deviate from the ground truth.

## 5.2 SUBJECTIVE EVALUATION

Tables 6 and 7 report results of our listening test. Respondents indicated a clear preference for the original music, confirming that human-composed music still outperforms generative approaches ($p < 0.05$ for all). The state-tuned model significantly improves over both the base and LoRA models ($p < 0.05$), with the most second place rankings. There was no significant difference between the base and LoRA model ($p > 0.05$). Further statistical analyses may be found in Appendix E.2.

Table 5: StyleRank evaluation of MIDI-RWKV generated content compared to GigaMIDI test set.

| Number of Bars ($N$) | Sets Closer to Corpus than Corpus to Corpus (of 100) | $p$-value |
|:---:|:---:|:---:|
| 2 | 52 | 0.764 |
| 4 | 47 | 0.617 |
| 8 | 42 | 0.089 |

Table 6: Subjective evaluation of MIDI-RWKV and finetunes on a POP909 test set.

| | Original music | Base model | LoRA $r = \alpha = 4$ | State tuned |
|:---|:---:|:---:|:---:|:---:|
| First place count ↑ | **54** | 27 | 23 | 36 |
| Second place count ↑ | 40 | 27 | 29 | **44** |
| Average rank ↓ | **2.057** | 2.779 | 2.807 | 2.357 |

Table 7: $p$-values for Table 6 after a Wilcoxon signed rank test with Holm-Bonferroni correction.

| | Original | Base model | LoRA $r = \alpha = 4$ | State tuned |
|:---|:---:|:---:|:---:|:---:|
| Original | - | - | - | - |
| Base model | $5.60 \cdot 10^{-5}$ | - | - | - |
| LoRA $r = \alpha = 4$ | $1.22 \cdot 10^{-5}$ | 0.896 | - | - |
| State tuned | 0.0486 | 0.0102 | 0.034 | - |

## 6 CONCLUSION

We presented MIDI-RWKV, a subquadratic foundation model for controllable and adaptable multi-track symbolic music infilling. Our approach addresses several limitations of existing music generation systems by reducing the complexity of the training objective and enabling effective style adaptation through state tuning. Objective and subjective experimental results demonstrate that MIDI-RWKV matches or outperforms comparable models, and that our proposed state tuning technique provides superior adaptability compared to LoRA approaches in the low-sample regime typical of individual composers and artists.

**Limitations.** While MIDI-RWKV aims to be style-agnostic and diverse in its outputs, it inherits from the biases of the data on which it was trained. Some musical styles and instruments are therefore overrepresented and others are underrepresented. We also train on relatively few attribute controls, which may lack the granularity that professional composers require.

Due to underoptimized RWKV inference pipelines and our decision not to implement inference tricks like caching hidden states, emulating random infilling using repeated single-section infilling takes slightly longer than with related systems. This difference is quantified in Appendix E.1, though we do not believe it is meaningful, as latencies of a few seconds were shown to be immaterial by Tchemeube et al. (2023). Nonetheless, we hope software engineering improvements can impress this issue.

While state tuning proved effective in Section 5.2, it may perform poorly on highly experimental or unusual compositional styles that deviate significantly from the distribution of the pretraining data, based on our rationale in Section 3.4, though this may be decreasingly important as music pretraining datasets scale to the giga and tera scales.

The system cannot yet compose in real time due to inference latency and a lack of ability to "stream" tokens, limiting its application in agentic and live contexts. This is in part due to the relative immaturity of the RWKV ecosystem compared to that of Transformers (Wolf et al., 2020), which also makes integration with mainstream systems and DAWs challenging. Future work will attempt to integrate MIDI-RWKV into DAWs, including Calliope (Tchemeube et al., 2025).

Our subjective evaluation is also limited: we did not give listeners the full 32 bars of context provided to the model, nor did we compare with the rank 32 LoRA or other models, as adding these would make the test prohibitively long. We hope future work can address some of these limitations.

ETHICAL STATEMENT

The subjective listening test of Section 5.2 was performed with the approval of an anonymized IRB. There are outstanding ethical and legal questions in computational music generation concerning the generation of music that exactly matches copyrighted work, and these questions extend far beyond the scope of this paper. State tuning also raises the risk of users finetuning the model to impersonate the styles of others, including potential copyrighted work.

REPRODUCIBILITY STATEMENT

We make our code and weights open-source and include a copy in the supplementary material, along with instructions in the README how to set them up to reproduce the majority of our experiments. In-depth descriptions of experimental procedures that were underspecified in the main text are also included in Appendix B.

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

## A    DESIGN OF SINGLE-SECTION INFILLING

Continuing from Section 3.3, single-section infilling was inspired by the limitations of other systems. While MIDI-GPT is technically able to infill arbitrary masking patterns in a single inference request, it performs multiple in practice, configured by inference hyperparameters $B$ and $T$. Each inference run infills one section of at most $B$ bars on each of at most $T$ tracks, where each section is contiguous on that track. For instance, $B = 2$, $T = 4$ by default in Calliope (Tchemeube et al., 2025), meaning that the model requires 4 requests to infill an 8-bar segment on one track, or 3 for a 6-bar segment on one track and a 3-bar segment on another.

However, MIDI-GPT is not aware of $B$ or $T$ during training, and is instead shown arbitrary masking patterns. This means that the prompts it sees in inference, which are limited by $B$ and $T$, constitute only a small subspace of the data it was shown during training, which have arbitrary $B$ and $T$. For instance, the model will have trained on many 5-track masking patterns, which would never appear during inference with default hyperparameters. This forces the model to model an unnecessarily high-dimensional space, or learn a harder task than it needs to.

We believe that the root cause of this issue is that the space of potential masking patterns increases exponentially with context length. The literature indicates that using arbitrary masking patterns at long training sequence lengths degrades performance, unless more long-context masking patterns are correspondingly added to the corpus to saturate the space. The Composer's Assistant models (Malandro, 2023; 2024) do the latter, achieving arbitrary masking patterns on up to about 16 measures (longer prompts typically require chunking) by using a nonuniform distribution of context sizes during training; MIDI-GPT uses the $B$ and $T$ hyperparameters to sidestep the problem by constraining the space of masking patterns to a tractable subspace at inference time.

We find neither solution satisfying, and instead propose to simplify the task by transforming arbitrary masking patterns into sequences of single-track, contiguous masking patterns during both training and inference (equivalent to $T = 1$, $B$ arbitrary in each training sample). This constitutes *single-section infilling*, allowing us to emulate arbitrary masking pattern infilling with a simpler training objective, shown on top in Figure 3 and in symbolic form in Figure 2. We propose single-section infilling as merely another potential vector—by no means the end-all method—of achieving the goal of truly arbitrary masking patterns over arbitrary contexts, which remains an open problem.

## B    DETAILS OF EXPERIMENTAL PROCEDURES

### B.1    TRAINING DATA PREPROCESSING

We tokenize our dataset into the representation described in Section 3.1, and discard files with less than 8 bars or 100 notes. The remaining files are shuffled and reprocesssed for each epoch. Since this adds a random component to data loading, we perform all experiments with a fixed seed of 42.

To create each example, we first randomly reorder the tracks so that the model learns different conditional orderings. From a randomly selected track $T$ of $L$ measures, we select a random section to infill of length $N = \max(choice([1, 2, 4, 8]), uniform(0.1, 0.4) \cdot L)$, where $choice$ and $uniform$ are as in the Python `random` module. We ensure that this section contains no empty measures; we found in preliminary experiments that the model learned from long, sparse tracks (e.g. chorus-only accompaniment) that the `Bar_None` bar separation token follows itself most frequently, resulting in mode collapse, so we avoid this in the infilling section during training. We still allow the model to see consecutive `Bar_None` tokens in the context to learn the dynamics of silence.

Next, we select $C$ bars on each side of the masked section across all tracks, such that $C$ is the greatest number of context bars available without exceeding the training sequence length; i.e. the $C$ selected bars encode to a representation (see Section 3.1) below the training sequence length, but selecting $C + 1$ bars on each side would encode to a representation in excess of the training sequence length. This results in a total window size of $N + 2C$ bars across $k$ tracks.

We then extract the contents of the $N$ infilling bars and replace them with $N$ `Infill_Bar` tokens to indicate to the model where the infilling content will go. We then compute the aforementioned attributes for each of the $N$ infilling bars and inject the attribute control tokens after the bar separation token starting each bar. Notably, this means all attribute controls are always inserted for each bar. The

block of attribute-controlled infilling bars is prefixed with a `FillBar_Start` token and suffixed with a `FillBar_End` token. The tracks are then concatenated together in a random order, followed by the infilled section, as in Figure 2. This is the training input to the model.

## B.2 INFERENCE

During inference, we receive as user input a score, a track ID $T$, a context length $C$, a list of attribute controls for each bar, and a range of $N$ bars to infill. We mask the $N$ specified bars on track $T$ and extract $C$ bars of context across all tracks before and after the infilling section. The tracks are then concatenated, and a `FillBar_Start` token and the attribute controls for the first bar are appended, to signal to the model to begin infilling. We sample until we have generated $N$ bars, at which point we stop and replace the bars in the original score.

The attribute control tokens for each bar are injected into the sequence after the previous bar is generated to condition the subsequent bar. These attribute controls can be provided by the user, computed from the original content, or a mixture of both. Because RWKV-7 is an RNN at inference time and thus has constant hidden state size, the window size $N + 2C$ is unlimited in inference, unlike training.

Because this is not possible with the standard HuggingFace Transformers (Wolf et al., 2020) sampling loop, we write our own custom sampling loop that is equivalent but also injects bar attribute controls into the sequence when a bar separation token is encountered. Our sampling loop allows for temperature, repetition penalty, top-k (Fan et al., 2018), and top-p (nucleus) (Holtzman et al., 2019) sampling, and can be extended to more. We also reject the sampling of two `Bar_None` tokens in succession, i.e. an empty measure, because this is rarely desirable—the user can make an empty measure themselves by simply doing nothing.

## B.3 STYLERANK

We adopt the same StyleRank setup as Pasquier et al. (2025), in particular defining "musical style" as the *stylistic characteristics delineated by a set of musical data* so as to avoid subjective definitions of "style". StyleRank measures the similarity of $k > 1$ groups of musical excerpts $\mathcal{G}_1, \ldots, \mathcal{G}_k$ to a style defined by a group of ground truth musical excerpts $\mathcal{C}$, using the cosine similarity in the embedding space of a random forest classifier trained to discriminate between the groups.

For each $N$ we test, we create $\mathcal{G}$, a set of 250 samples generated from the $N$-bar single-section infilling objective with $C = 4N$; $\mathcal{O}$, a set of 250 $N$-bar samples from the GigaMIDI test set, and $\mathcal{C}$, a set of 1000 $N$-bar samples from the GigaMIDI test set that is disjoint from $\mathcal{O}$. Let $\mathcal{S}_x$ denote a random subset of $x$ elements of $\mathcal{S}$. For each trial, we determine if the median similarity between $\mathcal{O}_{25}$ and $\mathcal{C}_{50}$ is less than the median similarity between $\mathcal{G}_{25}$ and $\mathcal{C}_{50}$, i.e. if the median similarity between two subsets of the corpus is less than or equal to the median similarity between a subset of the corpus and a set of generated excerpts. We count the number of trials for which the condition is true as the number of *sets closer to corpus than corpus to corpus*, as used in Table 5.

We collect the results for 100 trials and compute a binomial test. If there is no statistically significant difference between the count of trials for which the condition is true and the count for which the condition is false, we conclude that there is not a statistically significant difference between the generated material and the corpus with respect to the similarity metric we are using.

## B.4 SUBJECTIVE LISTENING TEST DESIGN

Each of the 28 participants of our subjective listening test was asked to describe their musical training as "Amateur - little to no formal schooling in music or music theory", "Intermediate - some formal schooling in music or music theory", or "Expert - work in industry or have a relevant degree". The split of respondents was 7/15/6 respectively, though we expect several applicants were hesitant to self-describe as "Expert"s.

Ten prompts were used for the evaluation, separated randomly into two sets of five prompts each, labeled A and B. Each participant was given the opposite set of the previous, as there was no particular ordering to the order of responses, resulting in 14 participants receiving the first set and 14 receiving the second set.

As comparing all models would take unreasonably long and might confound the results, the listening test only compared finetuned MIDI-RWKV outputs. For each prompt, participants were shown four clips in randomized order, one from each model tested, and were able to replay each at will. All clips were 8-bar infills with 32 bars of context from POP909, truncated to 4 bars around the infill region for rendering with the `symusic` library (Liao & Luo, 2024). They were then asked to rank the clips in order of *overall* preference; we eschew ratings at the advice of Yannakakis & Martínez (2015) and did not ask for rankings on any other metrics. The lengths of the clips ranged from 15 to 60 seconds.

## C  OBJECTIVE METRICS

**Content preservation.**  Introduced by Lu & Su (2018), the content preservation (sometimes referred to as *style preservation*) is calculated as follows. Each bar of the infilled content and the corresponding original content is split into $T$ time steps; we use $T = 16$. Each time step is associated with a pitch chroma vector $\vec{c}_t$ representing the probability distribution over the pitch classes (C, C#, D, . . . ) at that time. The moving average of the chroma vectors is calculated with a frame size of $T/2$ time steps, resulting in $TN$ averaged chroma vectors $\vec{a}_{o,t}$ for the original content and another $TN$ averaged chroma vectors $\vec{a}_{i,t}$ for the infilled content. The content preservation is then the average cosine similarity between contemporaneous chroma vectors across all timesteps,

$$CP(\{\vec{a_{o,t}}\}_{t=1}^{TN}, \{\vec{a_{i,t}}\}_{t=1}^{TN}) = \frac{1}{TN} \sum_{t=1}^{TN} \frac{\vec{a_{o,t}} \cdot \vec{a}_{i,t}}{\|\vec{a_{o,t}}\|\|\vec{a}_{i,t}\|}. \tag{1}$$

**Groove similarity.**  Introduced as *grooving pattern similarity* by Wu & Yang (2020), the groove similarity is calculated as follows. Let the grooving pattern $\vec{g}$ be a binary vector representing the positions in a bar at which there is at least one note onset. The groove similarity between the grooving patterns $\vec{g}_o$ of the original content and $\vec{g}_i$ of the infilled content is then the average number of onset positions (or absence of such) that match,

$$GS(\vec{g}_o, \vec{g}_i) = 1 - \frac{1}{\dim \vec{g}_o} \sum_{j=1}^{\dim \vec{g}_o - 1} \text{XOR}(g_{o,j}, g_{i,j}). \tag{2}$$

**Pitch class histogram entropy difference.**  Introduced by Wu & Yang (2020), the pitch class histogram entropy is calculated as follows. The pitch chroma vector $\vec{c}$ is calculated as in the calculation of content preservation, but is done across an entire bar instead of by time subdivisions. Its entropy is given by

$$\mathcal{H}(\vec{c}) = -\sum_{i=0}^{11} c_i \log c_i. \tag{3}$$

The pitch class histogram entropy *difference* is the difference between the pitch class histogram entropies of a bar of the original content and a bar of the infilled content.

**F1 score.**  A common measure in classification tasks, the F1 score is the harmonic mean of precision and recall, where precision is the ratio of correctly predicted notes to all predicted notes and recall is the ratio of correctly predicted notes to all ground truth notes. Formally, for the original content with note set $\mathcal{N}_o$ and the infilled content with note set $\mathcal{N}_i$, we have

$$F1(\mathcal{N}_o, \mathcal{N}_i) = 2 \cdot \frac{\text{precision} \cdot \text{recall}}{\text{precision} + \text{recall}} = \frac{2|\mathcal{N}_o \cap \mathcal{N}_i|}{|\mathcal{N}_o| + |\mathcal{N}_i|}. \tag{4}$$

## D  FURTHER STATE TUNING RESULTS

### D.1  STABILITY OF STATE TUNING

We perform the state tuning experiment detailed in Section 4.1 three times to establish that the high learning rate of 5e-2 still results in stable training dynamics, using a different 99/810 train/test split each time. The results are in Table 8. Stability is also demonstrated by the similarity between runs in Figure 5, particularly in the distribution of state magnitudes during inference in the middle plot.

Table 8: Single-section infilling evaluation of three state tuning runs on a POP909 test set.

|          | CP ↑            | GS ↑            | PCHE ↓          | F1 ↑            |
|----------|-----------------|-----------------|-----------------|-----------------|
| 2-bar v1 | $0.351 \pm 0.269$ | $0.952 \pm 0.037$ | $0.439 \pm 0.407$ | $0.073 \pm 0.190$ |
| 2-bar v2 | $0.332 \pm 0.250$ | $0.952 \pm 0.036$ | $0.447 \pm 0.377$ | $0.074 \pm 0.118$ |
| 2-bar v3 | $0.337 \pm 0.245$ | $0.947 \pm 0.043$ | $0.443 \pm 0.371$ | $0.066 \pm 0.108$ |
| 4-bar v1 | $0.345 \pm 0.152$ | $0.928 \pm 0.043$ | $0.355 \pm 0.312$ | $0.070 \pm 0.105$ |
| 4-bar v2 | $0.340 \pm 0.184$ | $0.924 \pm 0.036$ | $0.352 \pm 0.349$ | $0.073 \pm 0.126$ |
| 4-bar v3 | $0.332 \pm 0.152$ | $0.929 \pm 0.037$ | $0.346 \pm 0.378$ | $0.066 \pm 0.122$ |
| 8-bar v1 | $0.352 \pm 0.168$ | $0.908 \pm 0.065$ | $0.244 \pm 0.339$ | $0.054 \pm 0.107$ |
| 8-bar v2 | $0.350 \pm 0.154$ | $0.910 \pm 0.057$ | $0.246 \pm 0.172$ | $0.053 \pm 0.078$ |
| 8-bar v3 | $0.348 \pm 0.138$ | $0.909 \pm 0.043$ | $0.249 \pm 0.230$ | $0.056 \pm 0.096$ |

### D.2 State Tuning Interpretability

To provide justification for our claim in Section 3.4 that state tuning directs hidden state evolution to operate in a separate subspace of state space, we provide (to our knowledge) the first results interpreting the hidden state dynamics of a state-tuned RNN model in this section.

**Macro dynamics.** In Figure 5, we provide several comparisons of the three state-tuned models of Appendix D.1 (above) against the base model. 50 inference runs of the $N = 2, C = 8$ single-section infilling objective were performed with each model and the intermediate hidden states were logged and visualized.

We observe that the Frobenius distance between each state-tuned state matrix and the base model's state matrix is significantly higher than the per-step variance of the base model's state. Additionally, this distance does not decay significantly over time, indicating that the model's representations remain in a separate subspace of state space throughout inference.

The principal component analysis yields similar results: the hidden states of the v2 and v3 state tuned models are distributed so closely in the PC1/2 graph that most v2 points are obscured by v3 points, and both distributions lie far from the base model's distribution. While the states of the v1 state tuned model lie in a separate region, they are still far from the base model, and qualitatively the distribution looks similar to those of the v2 and v3 models (long and skinny with some kinks in the middle).

The PC3/4 projection shows the v1 states overlapping with the base model significantly, while the distribution of v2 and v3 states lie approximately equidistant in opposite directions along PC3. The PC5/6 projection shows very little variance between models, with outlying clusters each corresponding to individual prompts; we believe this shows prompt-specific variations being captured in higher-order components. Together, these projections indicate that state tuning has effectively biased the hidden state evolution of the base model to operate in a separate subspace of state space, as claimed.

**Micro dynamics.** We also visualize individual model trajectories to analyze fine-grained trajectory structure. Figures 6 and 7 show the hidden state evolution for a single $N = 2, C = 8$ and $N = 8, C = 32$ single-section infilling prompt, respectively, across the base model and all three state-tuned models of Appendix D.1. 6-dimensional PCA was performed on all models' collective hidden states before focusing each graph on a local subspace of the overall PCA space.

These examples reinforce some findings about the structure of state subspaces suggested by Figure 5. In both figures, the state tuned v2 and v3 models have very similar trajectories in PC1/2, and their trajectories in PC3/4 are approximately mirrored along PC3. The PC5/6 trajectories are also almost identical for all models in the $N = 2, C = 8$ prompt of Figure 6, diverging slightly more in the longer $N = 8, C = 32$ prompt of Figure 7, although they maintain similar structures of "clumps" on the right side with five or so "fingers" extending out to the left.

We also observe some individual trends. Notably, all trajectories begin far from their respective model's typical operating region (indicated by distance from initial state to the bulk of the trajectory), particularly in PC1/2, suggesting initialization plays a significant role in early generation dynamics.

Table 9: Single-section infilling evaluation of MIDI-RWKV and MIDI-GPT finetunes on a POP909 test set. "MIDI-" prefixes are removed. S refers to state tuning, T token parity, W wall-clock parity.

| Model | CP ↑ | GS ↑ | PCHE ↓ | F1 ↑ |
|---|---|---|---|---|
| 2-bar RWKV | $0.316 \pm 0.182$ | $0.947 \pm 0.039$ | $0.497 \pm 0.415$ | $0.063 \pm 0.146$ |
| 2-bar RWKV S | $\mathbf{0.351 \pm 0.269}$ | $\mathbf{0.952 \pm 0.037}$ | $\mathbf{0.439 \pm 0.407}$ | $\mathbf{0.073 \pm 0.190}$ |
| 2-bar GPT | $0.260 \pm 0.254$ | $0.948 \pm 0.045$ | $0.577 \pm 0.405$ | $0.021 \pm 0.026$ |
| 2-bar GPT T | $0.269 \pm 0.206$ | $0.946 \pm 0.396$ | $0.531 \pm 0.282$ | $0.017 \pm 0.042$ |
| 2-bar GPT W | $0.256 \pm 0.217$ | $0.948 \pm 0.045$ | $0.517 \pm 0.381$ | $0.043 \pm 0.037$ |
| 4-bar RWKV | $0.293 \pm 0.159$ | $0.925 \pm 0.052$ | $0.433 \pm 0.348$ | $\mathbf{0.072 \pm 0.107}$ |
| 4-bar RWKV S | $\mathbf{0.345 \pm 0.152}$ | $0.928 \pm 0.043$ | $\mathbf{0.355 \pm 0.312}$ | $0.070 \pm 0.105$ |
| 4-bar GPT | $0.266 \pm 0.275$ | $0.918 \pm 0.059$ | $0.532 \pm 0.421$ | $0.015 \pm 0.027$ |
| 4-bar GPT T | $0.286 \pm 0.253$ | $0.910 \pm 0.555$ | $0.485 \pm 0.272$ | $0.052 \pm 0.082$ |
| 4-bar GPT W | $0.336 \pm 0.209$ | $\mathbf{0.929 \pm 0.048}$ | $0.449 \pm 0.042$ | $0.026 \pm 0.021$ |
| 8-bar RWKV | $0.287 \pm 0.116$ | $0.884 \pm 0.062$ | $0.314 \pm 0.276$ | $\mathbf{0.054 \pm 0.107}$ |
| 8-bar RWKV S | $\mathbf{0.352 \pm 0.168}$ | $0.908 \pm 0.065$ | $\mathbf{0.244 \pm 0.339}$ | $\mathbf{0.054 \pm 0.072}$ |
| 8-bar GPT | $0.284 \pm 0.263$ | $\mathbf{0.938 \pm 0.051}$ | $0.465 \pm 0.712$ | $0.008 \pm 0.014$ |
| 8-bar GPT T | $0.283 \pm 0.233$ | $0.891 \pm 0.047$ | $0.413 \pm 0.201$ | $0.035 \pm 0.050$ |
| 8-bar GPT W | $0.330 \pm 0.223$ | $0.911 \pm 0.058$ | $0.384 \pm 0.232$ | $0.034 \pm 0.041$ |

Furthermore, all state-tuned models seem to exhibit high-level similarities in their trajectory shapes, despite their trajectories being far apart in PCA space. In addition to the striking similarity between state-tuned v2 and v3 model trajectories in PC1/2, the v1 trajectory also looks similar when reflected across the antidiagonal in both figures. All models also exhibit similar local dynamics in PC3/4: in Figure 6, all have a tall "finger" extending up into positive PC4 before returning to negative values; in Figure 7, all models tend to oscillate along PC3 while increasing along PC4. We think this reflects the model's ability to retain dynamics that correspond to overall musical knowledge even when operating in different subspaces of state space.

**Training dynamics.**  Figures 8 and 9 show the state dynamics over the course of the finetuning process. Checkpoints were saved every 2 epochs during state tuning, and the procedure of the "Macro dynamics" subsection above was reused to collect hidden states on $N = 2, C = 8$ (Figure 8) and $N = 8, C = 32$ (Figure 9) single-section infilling objectives, where PCA was then performed on the aggregate hidden states to create the visualizations shown.

We observe that the trajectories are rather straight in PC1/2. Notably, the learning rate was held constant during training at 5e-2, suggesting that the apparent convergence of hidden states is due to the rapid and effective discovery of local minima (as it cannot be due to learning rate decay). This suggests that the convergence of the state-tuned v2 and v3 models is due to them both finding the same local minimum despite being trained on different subsets of POP909, an interesting discovery.

As expected, the trajectories in PC5/6 are broadly incomprehensible, though we note that all state-tuned hidden states lie separate from the base model's hidden states in the $N = 2, C = 8$ objective. We also see that the hidden states tend to stabilize fairly rapidly in PC3/4, finding the general subspace in which they remain after only 6 or so epochs (equating to only a few hundred training samples). This is especially apparent on the $N = 8, C = 32$ objective, where hidden state samples from different epochs are mixed within each model's cluster.

D.3 COMPARISONS TO MIDI-GPT LoRA

We conducted finetuning experiments on MIDI-GPT using LoRA $r = \alpha = 4$, based on parity with our RWKV LoRA $r = \alpha = 4$ model from Section 4.1 in 1) wall-clock time and 2) number of tokens (as wall-clock time is not always comparable). We did not compare against Composer's Assistant as its finetuning method uses full-parameter finetuning, which we thought would be an improper comparison. Table 9 contains the results; we omit the RWKV LoRA results to not clutter the table.

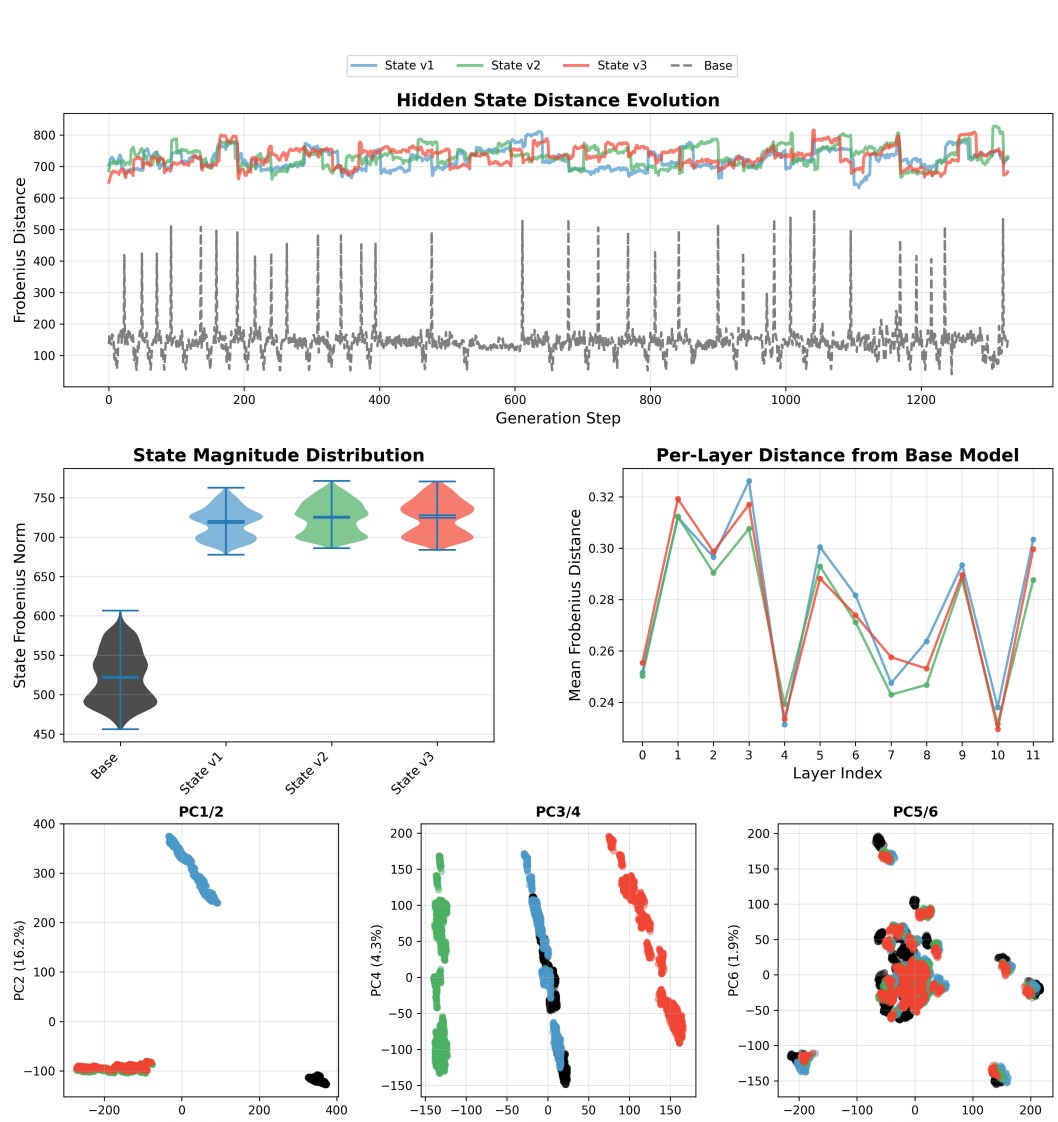

Figure 5: Visualizations of hidden state dynamics with state tuning on $N = 2, C = 8$ objectives.
**Top:** Frobenius distance between state-tuned and base model states at each timestep, plus baseline of the base model's state distance to its previous state.
**Middle left:** Violin plot of state magnitudes across the hidden state samples taken.
**Middle right:** Layer-wise Frobenius distance between state-tuned and base model states.
**Bottom:** Three 2-dimensional principal component analyses of each model's hidden states along the first six principal components. State v2 model data points are obscured by overlapping v3 data points in the bottom left, and likewise for base model data points by v1 data points in the bottom center.

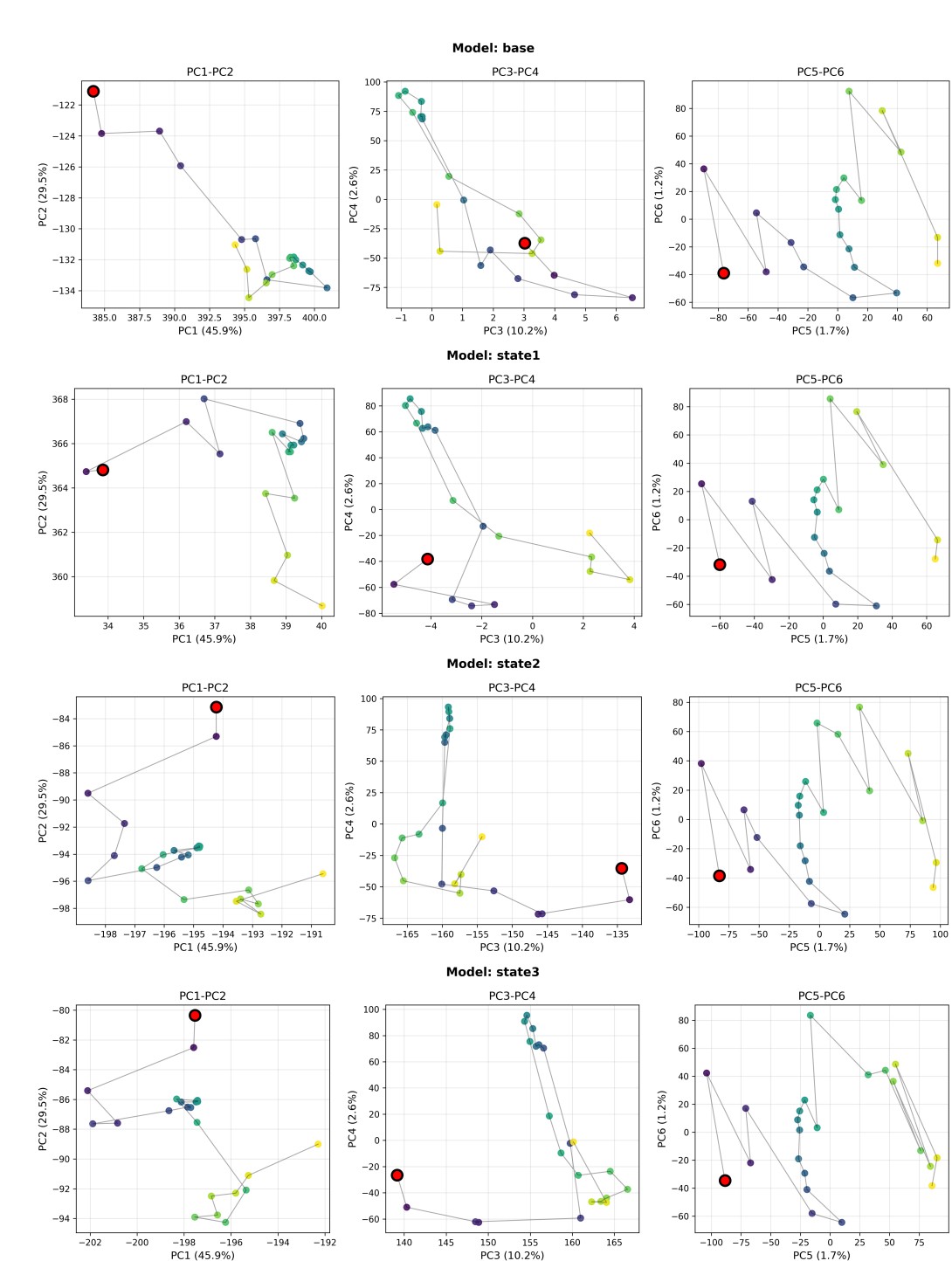

Figure 6: Hidden state trajectories of each model in its local subspace of the first six dimensions of PCA space, on a single $N = 2, C = 8$ single-section infilling prompt. PCA is performed on the aggregate of all four models' hidden states, and each graph is zoomed in on that model's local trajectory space (note the different axis tick marks). The color gradient moves from dark to bright as the trajectory progresses, and the red circle indicates the initial state.

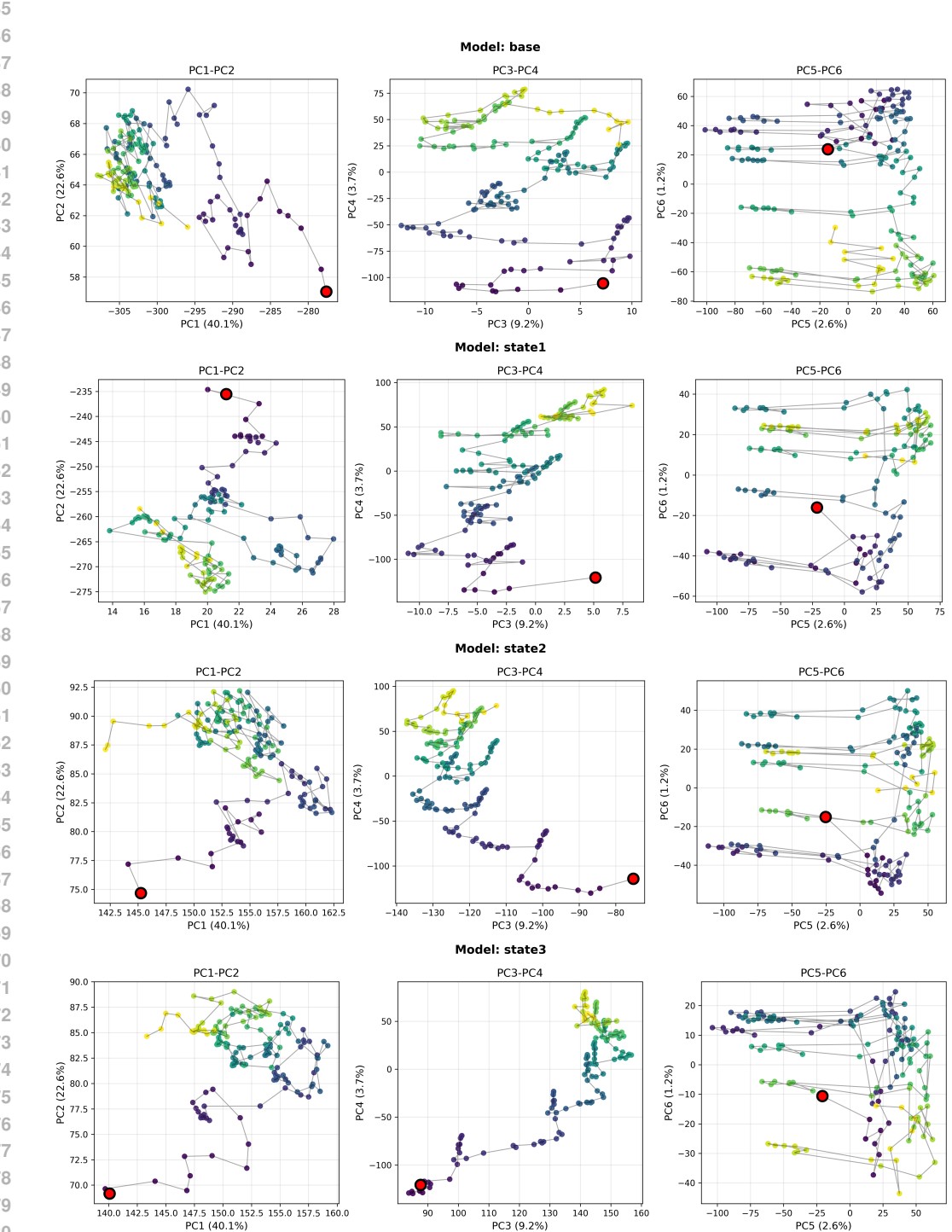

Figure 7: Hidden state trajectories of each model in its local subspace of the first six dimensions of PCA space, on a single $N = 8, C = 32$ single-section infilling prompt. PCA is performed on the aggregate of all four models' hidden states, and each graph is zoomed in on that model's local trajectory space (note the different axis tick marks). The color gradient moves from dark to bright as the trajectory progresses, and the red circle indicates the initial state.

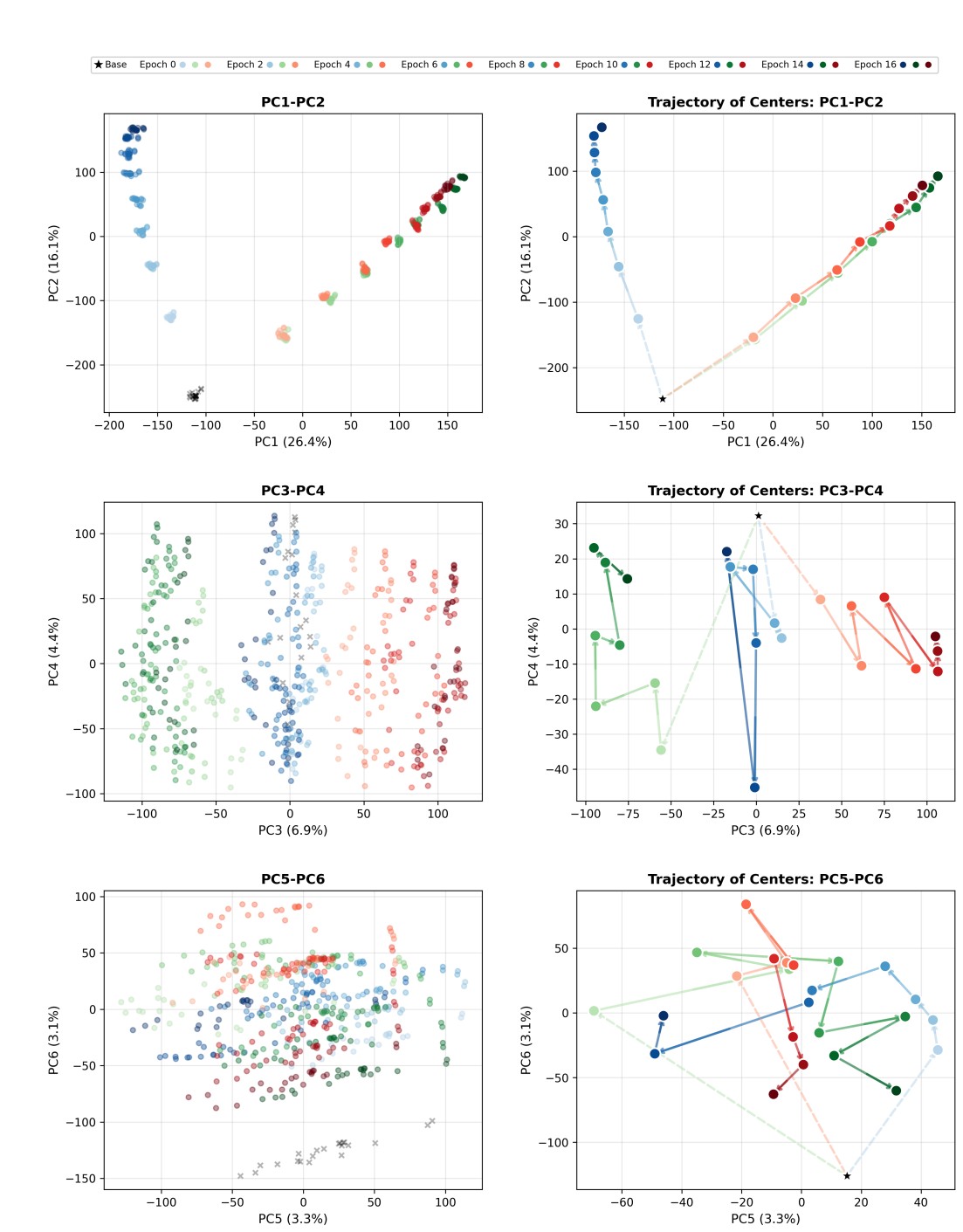

Figure 8: Hidden state PCA of each state-tuned model over the course of training on $N = 2, C = 8$ objectives. Left shows the full plots while right shows only the centers. The v1 model is shown in blue, v2 in green, and v3 in red, where the shades move from light to dark as training progresses.

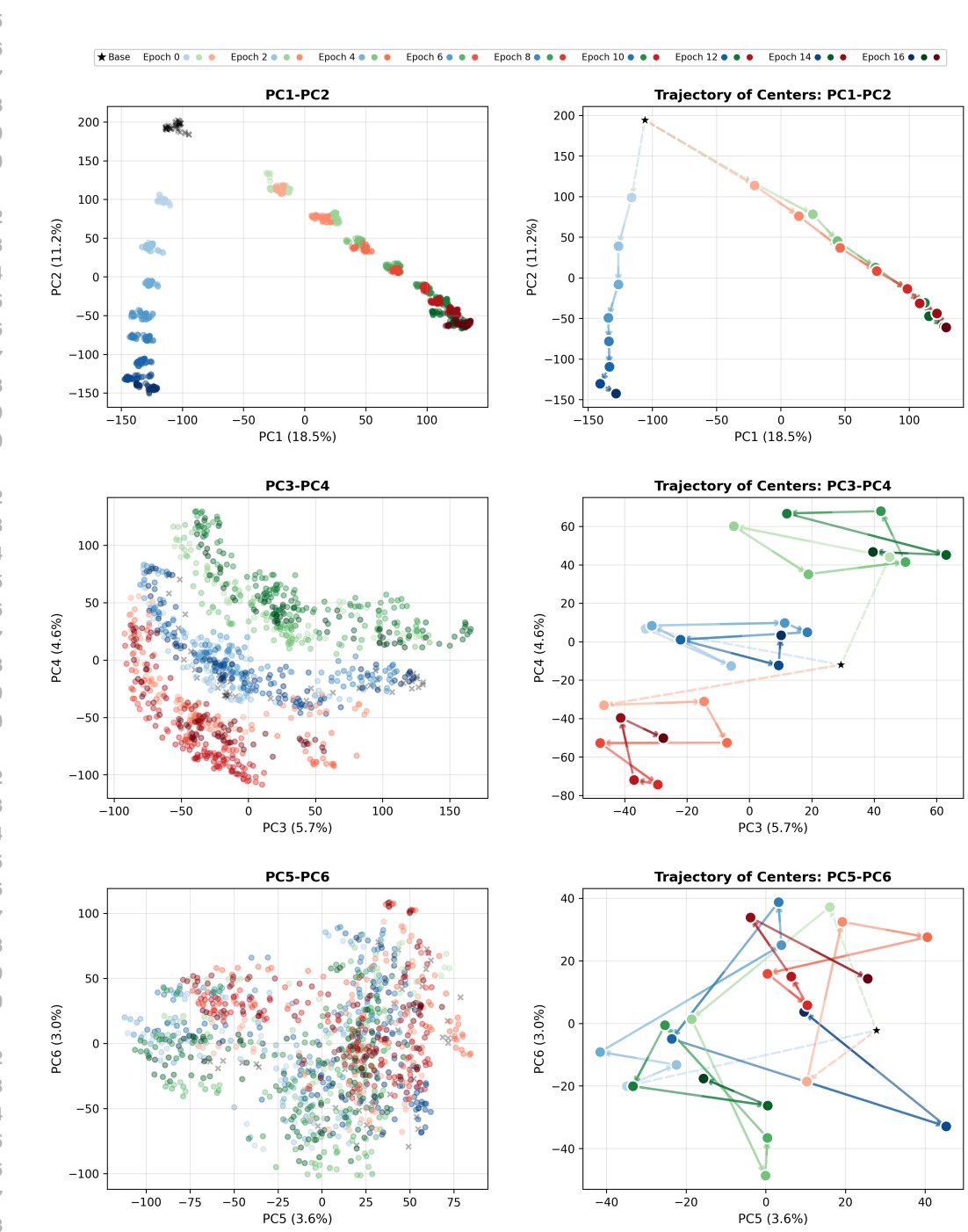

Figure 9: Hidden state PCA of each state-tuned model over the course of training on $N = 8, C = 32$ objectives. Left shows the full plots while right shows only the centers. The v1 model is shown in blue, v2 in green, and v3 in red, where the shades move from light to dark as training progresses.

Table 10: Single-section infilling evaluation of MIDI-RWKV finetunes on a Pop1K7 test set.

| | CP ↑ | GS ↑ | PCHE ↓ | F1 ↑ |
|---|---|---|---|---|
| 2-bar base | $0.566 \pm 0.157^a$ | $0.905 \pm 0.054^a$ | $0.320 \pm 0.272^a$ | $0.019 \pm 0.043^a$ |
| 2-bar LoRA 4 | $0.568 \pm 0.201^a$ | $\mathbf{0.914 \pm 0.056}^b$ | $0.317 \pm 0.259^a$ | $\mathbf{0.023 \pm 0.036}^a$ |
| 2-bar LoRA 32 | $0.563 \pm 0.173^a$ | $0.908 \pm 0.043^a$ | $0.312 \pm 0.266^b$ | $0.021 \pm 0.033^a$ |
| 2-bar state | $\mathbf{0.575 \pm 0.158}^b$ | $0.912 \pm 0.050^b$ | $\mathbf{0.295 \pm 0.245}^c$ | $0.018 \pm 0.041^a$ |
| 4-bar base | $0.508 \pm 0.136^a$ | $0.856 \pm 0.071^a$ | $0.279 \pm 0.234^a$ | $0.027 \pm 0.032^a$ |
| 4-bar LoRA 4 | $0.537 \pm 0.127^b$ | $0.858 \pm 0.069^a$ | $0.248 \pm 0.219^b$ | $0.027 \pm 0.033^a$ |
| 4-bar LoRA 32 | $0.536 \pm 0.122^b$ | $0.854 \pm 0.066^a$ | $0.251 \pm 0.205^c$ | $\mathbf{0.030 \pm 0.032}^b$ |
| 4-bar state | $\mathbf{0.545 \pm 0.132}^c$ | $\mathbf{0.875 \pm 0.074}^b$ | $\mathbf{0.247 \pm 0.194}^b$ | $0.029 \pm 0.041^b$ |
| 8-bar base | $0.516 \pm 0.133^a$ | $0.824 \pm 0.058^a$ | $0.303 \pm 0.237^a$ | $0.033 \pm 0.036^a$ |
| 8-bar LoRA 4 | $0.526 \pm 0.123^b$ | $0.815 \pm 0.077^b$ | $0.258 \pm 0.210^b$ | $0.34 \pm 0.057^a$ |
| 8-bar LoRA 32 | $0.521 \pm 0.127^c$ | $0.825 \pm 0.072^a$ | $\mathbf{0.215 \pm 0.197}^c$ | $\mathbf{0.035 \pm 0.066}^a$ |
| 8-bar state | $\mathbf{0.552 \pm 0.130}^d$ | $\mathbf{0.849 \pm 0.076}^c$ | $0.219 \pm 0.203^c$ | $0.032 \pm 0.031^a$ |

### D.4 EXPERIMENTS ON FURTHER DATASETS

We provide here state tuning results (in addition to those of Table 4) on the following datasets:

- Pop1K7 (Hsiao et al., 2021), comprising 1747 pop piano performances (Table 10);
- YM2413-MDB (Choi et al., 2022), comprising 669 80s FM videogame tracks (Table 11);
- the Nottingham dataset (Allwright, 2003), comprising 1037 folk songs (Table 12); and
- the `waltzes` subset of the Nottingham dataset, comprising 35 waltzes (Table 13).

The methodology was the same as described in Section 4.1, using a 10/90 train/test split, except that we did not restrict the models to melody only. For the Nottingham `waltzes` subset, we trained for 64 epochs instead of 16 to increase the quantity of gradient updates to a reasonable number, as each epoch contained only 4 samples. Since learning rate was held constant, this only affected number of tokens seen during training. The `midi_analyzed` set from Pop1K7 and the `adjust_tempo_remove_delayed_inst` set from YM2413-MDB were used for the respective datasets, being the most refined versions of each.

As before, different superscripts in the same column and section indicate statistically significant ($p < 0.05$) differences. Results are reported in Tables 10–13. We again see generally significant improvements on all datasets with state tuning over the alternatives, even on the Nottingham `waltzes` subset where there were only 4 source MIDI files (which comprised many more training samples—see Appendix B.1), demonstrating that state tuning can be effective even in the low-sample regime.

## E FURTHER GENERAL RESULTS

### E.1 INFERENCE LATENCY COMPARISON WITH COMPOSER'S ASSISTANT

Because RWKV-7 and other recurrent models have not benefited from the years of architecture-specific optimizations Transformer models enjoy, they remain slower on short input lengths despite their asymptotic improvements. Furthermore, the serial dependency introduced by single-section infilling requires multiple inference calls to infill a single section, further slowing inference. We do not implement methods to expedite inference, such as reusing prefix states to avoid unnecessary recomputation, resulting in our model being considerably slower than Composer's Assistant at inference time.

This difference is quantified in Figure 10, where we compare the latency of Composer's Assistant vs. MIDI-RWKV on 100 8- and 16-bar random infilling prompts (the natural format of Composer's Assistant). Experiments were performed at batch size 1 on an Intel i9-14900K CPU, as our target audience will largely be using CPU inference.

Table 11: Single-section infilling evaluation of MIDI-RWKV finetunes on a YM2413-MDB test set.

| | CP ↑ | GS ↑ | PCHE ↓ | F1 ↑ |
|---|---|---|---|---|
| 2-bar base | $0.409 \pm 0.237^a$ | $0.882 \pm 0.062^a$ | $0.715 \pm 0.588^a$ | $\mathbf{0.062 \pm 0.099}^a$ |
| 2-bar LoRA 4 | $0.405 \pm 0.245^a$ | $0.864 \pm 0.096^a$ | $0.675 \pm 0.596^b$ | $0.057 \pm 0.123^a$ |
| 2-bar LoRA 32 | $0.416 \pm 0.232^b$ | $0.872 \pm 0.081^b$ | $0.617 \pm 0.574^c$ | $0.055 \pm 0.144^a$ |
| 2-bar state | $\mathbf{0.430 \pm 0.221}^c$ | $\mathbf{0.885 \pm 0.084}^c$ | $\mathbf{0.585 \pm 0.523}^d$ | $0.058 \pm 0.159^a$ |
| 4-bar base | $0.391 \pm 0.201^a$ | $0.807 \pm 0.093^a$ | $0.600 \pm 0.509^a$ | $\mathbf{0.068 \pm 0.148}^a$ |
| 4-bar LoRA 4 | $0.375 \pm 0.203^b$ | $0.812 \pm 0.104^b$ | $0.604 \pm 0.531^a$ | $0.065 \pm 0.100^a$ |
| 4-bar LoRA 32 | $0.392 \pm 0.197^a$ | $0.828 \pm 0.088^c$ | $0.586 \pm 0.512^b$ | $\mathbf{0.068 \pm 0.124}^a$ |
| 4-bar state | $\mathbf{0.401 \pm 0.206}^c$ | $\mathbf{0.841 \pm 0.097}^d$ | $\mathbf{0.540 \pm 0.497}^c$ | $0.053 \pm 0.123^b$ |
| 8-bar base | $0.408 \pm 0.159^a$ | $\mathbf{0.828 \pm 0.101}^a$ | $0.637 \pm 0.551^a$ | $0.037 \pm 0.103^a$ |
| 8-bar LoRA 4 | $0.422 \pm 0.154^b$ | $0.786 \pm 0.117^b$ | $0.595 \pm 0.540^b$ | $\mathbf{0.045 \pm 0.110}^b$ |
| 8-bar LoRA 32 | $0.437 \pm 0.165^c$ | $0.790 \pm 0.098^b$ | $0.580 \pm 0.538^c$ | $0.042 \pm 0.095^b$ |
| 8-bar state | $\mathbf{0.459 \pm 0.161}^d$ | $0.797 \pm 0.110^c$ | $\mathbf{0.571 \pm 0.544}^d$ | $0.043 \pm 0.091^b$ |

Table 12: Single-section infilling evaluation of MIDI-RWKV finetunes on a Nottingham test set.

| | CP ↑ | GS ↑ | PCHE ↓ | F1 ↑ |
|---|---|---|---|---|
| 2-bar base | $0.531 \pm 0.279^a$ | $0.963 \pm 0.028^a$ | $0.342 \pm 0.313^a$ | $0.269 \pm 0.228^a$ |
| 2-bar LoRA 4 | $0.577 \pm 0.294^b$ | $0.964 \pm 0.027^a$ | $0.303 \pm 0.296^b$ | $0.301 \pm 0.260^b$ |
| 2-bar LoRA 32 | $0.586 \pm 0.278^b$ | $\mathbf{0.965 \pm 0.026}^a$ | $0.311 \pm 0.312^b$ | $\mathbf{0.332 \pm 0.301}^c$ |
| 2-bar state | $\mathbf{0.596 \pm 0.255}^c$ | $0.965 \pm 0.027^a$ | $\mathbf{0.286 \pm 0.292}^c$ | $0.315 \pm 0.288^b$ |
| 4-bar base | $0.509 \pm 0.367^a$ | $0.956 \pm 0.032^a$ | $0.294 \pm 0.256^a$ | $0.147 \pm 0.125^a$ |
| 4-bar LoRA 4 | $0.511 \pm 0.349^a$ | $0.959 \pm 0.031^b$ | $0.317 \pm 0.297^b$ | $\mathbf{0.163 \pm 0.176}^b$ |
| 4-bar LoRA 32 | $0.530 \pm 0.377^b$ | $0.958 \pm 0.028^b$ | $\mathbf{0.290 \pm 0.283}^c$ | $0.154 \pm 0.141^c$ |
| 4-bar state | $\mathbf{0.542 \pm 0.375}^c$ | $\mathbf{0.961 \pm 0.030}^b$ | $0.292 \pm 0.312^c$ | $0.154 \pm 0.133^c$ |
| 8-bar base | $0.558 \pm 0.315^a$ | $0.958 \pm 0.028^a$ | $0.142 \pm 0.148^a$ | $\mathbf{0.087 \pm 0.118}^a$ |
| 8-bar LoRA 4 | $0.556 \pm 0.299^a$ | $0.958 \pm 0.031^a$ | $0.166 \pm 0.132^b$ | $0.085 \pm 0.094^a$ |
| 8-bar LoRA 32 | $0.698 \pm 0.267^b$ | $0.977 \pm 0.018^b$ | $0.114 \pm 0.167^c$ | $0.069 \pm 0.082^b$ |
| 8-bar state | $\mathbf{0.721 \pm 0.348}^b$ | $\mathbf{0.984 \pm 0.025}^c$ | $\mathbf{0.101 \pm 0.125}^d$ | $0.062 \pm 0.097^b$ |

Table 13: Single-section infilling evaluation of MIDI-RWKV finetunes on a Nottingham waltz set.

| | CP ↑ | GS ↑ | PCHE ↓ | F1 ↑ |
|---|---|---|---|---|
| 2-bar base | $0.474 \pm 0.314^a$ | $0.959 \pm 0.014^a$ | $0.377 \pm 0.320^a$ | $0.291 \pm 0.285^a$ |
| 2-bar LoRA 4 | $0.507 \pm 0.302^b$ | $0.961 \pm 0.015^a$ | $0.328 \pm 0.292^b$ | $\mathbf{0.339 \pm 0.297}^b$ |
| 2-bar LoRA 32 | $0.521 \pm 0.308^c$ | $0.964 \pm 0.012^b$ | $0.362 \pm 0.331^a$ | $0.298 \pm 0.273^a$ |
| 2-bar state | $\mathbf{0.559 \pm 0.287}^d$ | $\mathbf{0.966 \pm 0.021}^c$ | $\mathbf{0.303 \pm 0.281}^c$ | $0.316 \pm 0.300^b$ |
| 4-bar base | $0.351 \pm 0.357^a$ | $0.970 \pm 0.016^a$ | $0.253 \pm 0.208^a$ | $0.187 \pm 0.157^a$ |
| 4-bar LoRA 4 | $0.383 \pm 0.324^b$ | $0.969 \pm 0.014^a$ | $0.212 \pm 0.193^b$ | $0.182 \pm 0.159^a$ |
| 4-bar LoRA 32 | $0.374 \pm 0.344^c$ | $0.970 \pm 0.021^a$ | $0.218 \pm 0.186^b$ | $0.230 \pm 0.154^b$ |
| 4-bar state | $\mathbf{0.392 \pm 0.341}^d$ | $\mathbf{0.972 \pm 0.020}^b$ | $\mathbf{0.156 \pm 0.199}^c$ | $\mathbf{0.250 \pm 0.186}^c$ |
| 8-bar base | $0.681 \pm 0.246^a$ | $0.973 \pm 0.022^a$ | $0.090 \pm 0.095^a$ | $0.029 \pm 0.029^a$ |
| 8-bar LoRA 4 | $0.706 \pm 0.319^b$ | $0.973 \pm 0.020^a$ | $0.092 \pm 0.146^a$ | $0.035 \pm 0.033^b$ |
| 8-bar LoRA 32 | $0.705 \pm 0.237^b$ | $0.985 \pm 0.015^b$ | $0.086 \pm 0.102^b$ | $0.038 \pm 0.028^b$ |
| 8-bar state | $\mathbf{0.716 \pm 0.255}^c$ | $\mathbf{0.994 \pm 0.008}^c$ | $\mathbf{0.074 \pm 0.085}^c$ | $\mathbf{0.039 \pm 0.032}^b$ |

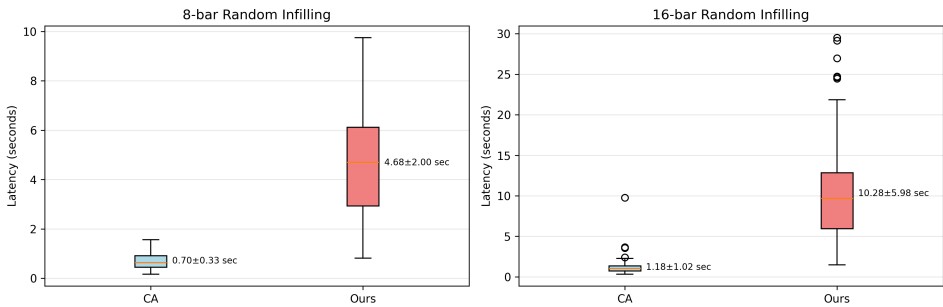

Figure 10: Comparison of inference latency of Composer's Assistant and MIDI-RWKV on random infilling objectives, in seconds.

Table 14: Kendall's $W$ for each musical sample in the subjective evaluation. Samples with statistically significant rater agreement ($p < 0.05$) are marked with *.

| Set | Question No. | Kendall's $W$ | $\chi^2$ | $p$-value |
|-----|--------------|---------------|----------|-----------|
| A | 1 | 0.208 | 8.74 | 0.033* |
| A | 2 | 0.251 | 10.54 | 0.015* |
| A | 3 | 0.163 | 6.86 | 0.077 |
| A | 4 | 0.231 | 9.69 | 0.021* |
| A | 5 | 0.210 | 8.83 | 0.032* |
| B | 1 | 0.214 | 9.00 | 0.029* |
| B | 2 | 0.045 | 1.89 | 0.597 |
| B | 3 | 0.063 | 2.66 | 0.448 |
| B | 4 | 0.055 | 2.31 | 0.510 |
| B | 5 | 0.210 | 8.83 | 0.032* |
| Mean | | 0.165 | 6.93 | – |

We observe a significant increase in inference latency, but we believe that even latency on the order of 10 seconds for a 16-bar request is still reasonable, as similar inference times still proved negligible for composers in the study of Tchemeube et al. (2023) on MMM. Additionally, we intend to speed up inference as we continue our integration with Calliope.

### E.2  FURTHER STATISTICAL TESTS ON SUBJECTIVE RESULTS

In addition to the Wilcoxon signed-rank test in Table 7, here we provide further statistical analysis of the subjective evaluation results. A Friedman test across all 140 evaluations established that there are significant differences between model outputs according to respondents ($\chi^2 = 32.63$, $p < 0.001$).

To assess inter-rater agreement, we computed Kendall's coefficient of concordance $W$ (Kendall & Smith, 1939) for each question. Agreement varied considerably by sample (Table 14), with 6 of 10 samples showing significant rater consensus ($p < 0.05$) with moderate agreement levels on the order of 0.2. We believe the moderate agreement level reflects the inherently subjective nature of music quality evaluation, while the Friedman test above and pairwise comparisons of Table 7 demonstrate that consistent quality differences indeed exist across models in our subjective evaluation results.

### E.3  SAMPLING PARAMETER ABLATIONS

We compare objective metrics on different sampling parameters in Table 15. The default parameters, as suggested by Rizzotti (2025), are temperature=1.0, repetition penalty=1.2, top-k=20, and top-p=0.95. One parameter was varied at a time to other common values to determine their impact on generation, and 100 examples were generated with each. We observed that, while the default choice of sampling parameters does not achieve the best score in all categories, it performs admirably well

Table 15: Objective evaluations under different sampling parameters.

| | CP ↑ | GS ↑ | PCHE ↓ | F1 ↑ |
|---|---|---|---|---|
| Default | $0.419 \pm 0.225$ | $\mathbf{0.928 \pm 0.077}$ | $0.466 \pm 0.423$ | $0.093 \pm 0.146$ |
| Temperature 0.8 | $\mathbf{0.438 \pm 0.226}$ | $0.915 \pm 0.072$ | $0.490 \pm 0.462$ | $0.115 \pm 0.226$ |
| Temperature 1.2 | $0.358 \pm 0.188$ | $0.895 \pm 0.074$ | $0.501 \pm 0.553$ | $0.078 \pm 0.096$ |
| Rep. penalty 1.0 | $0.382 \pm 0.228$ | $0.924 \pm 0.061$ | $0.586 \pm 0.554$ | $\mathbf{0.125 \pm 0.248}$ |
| Rep. penalty 1.4 | $0.397 \pm 0.202$ | $0.895 \pm 0.088$ | $\mathbf{0.423 \pm 0.375}$ | $0.075 \pm 0.092$ |
| Top-p 0.9 | $0.416 \pm 0.224$ | $0.919 \pm 0.070$ | $0.476 \pm 0.486$ | $0.066 \pm 0.079$ |
| Top-p 0.98 | $0.395 \pm 0.209$ | $0.901 \pm 0.084$ | $0.503 \pm 0.449$ | $0.063 \pm 0.080$ |
| Top-k 15 | $0.419 \pm 0.229$ | $0.896 \pm 0.079$ | $0.547 \pm 0.587$ | $0.088 \pm 0.158$ |
| Top-k 30 | $0.391 \pm 0.245$ | $0.910 \pm 0.078$ | $0.449 \pm 0.447$ | $0.075 \pm 0.122$ |

in each. Please note that these ablations were performed on a checkpoint from an earlier training run than the checkpoint used in the main body experiments, which is why the numbers do not match.

There are several subjective reasons for the choice of default sampling parameters: we found that a temperature below 1.0 caused the model to repeat the same output several times, which will inevitably bring frustration from the user but is not captured in the objective metrics. Generations with a repetition penalty above 1.2 also tended to lose structural coherence, likely because the high penalty discouraged repetitions that are natural in music; this is reflected in the groove similarity scores.

### E.4 FINETUNED ATTRIBUTE CONTROL ADHERENCE EVALUATIONS

We provide attribute control evaluations on the base model, LoRA-trained models with $r = \alpha = 4$ and $r = \alpha = 32$, and a state tuned representative model. The left subfigure of each is the average absolute difference between real and intended note density; the right is the success rate of categorical control tokens. Across Figures 11, 12, and 13, the state tuned model typically outperforms the base model slightly, which in turn outperforms the LoRA finetuned models.

### E.5 FURTHER OBJECTIVE COMPARISONS

We provide here two comparisons that did not fit in the main text: against Composer's Assistant 2 (CA2) (Malandro, 2024), a model almost 5.5 times larger than MIDI-RWKV by parameter count, and against Polyffusion (Min et al., 2023), a single-track-only inpainting model. We provide the former comparison to contextualize our model's performance within the broader multi-track infilling field, and the latter to demonstrate that multi-track infilling can outperform single-track infilling even on single-track tasks.

**Composer's Assistant 2.** The comparison against CA2 was performed on both the 8- and 16-bar random infilling objectives defined in Section 4.2, which Malandro (2023; 2024) evaluated on natively. Inference was performed using the official code and weights from the Composer's Assistant 2 paper. Results are reported in Table16 and compared against our random infilling results of Table 3.

As expected, CA2 performs better than MIDI-RWKV on almost all metrics, although curiously we still outperform it in the content preservation metric. We believe this is explained by the stark difference in F1 scores: CA2 (at least with its default sampling parameters) is better at regenerating the exact ground truth, indicated by very high F1 scores, while MIDI-RWKV can approximate the ground truth better without reproducing it verbatim.

**Polyffusion.** The comparison against Polyffusion was done using the $N = 2, 4, 8$ single-section infilling objectives, which Polyffusion most nearly supports, on the same POP909 test set we used for the state tuning experiment of Section 4.2 and Table 4, to allow us to reuse our evaluation data for MIDI-RWKV. Inference was performed using the official code and `sdf_chd8bar` weights from the Polyffusion paper, as these were the only public weights we could find. The results are reported in Table 17 and compared against the base MIDI-RWKV model results of Table 4.

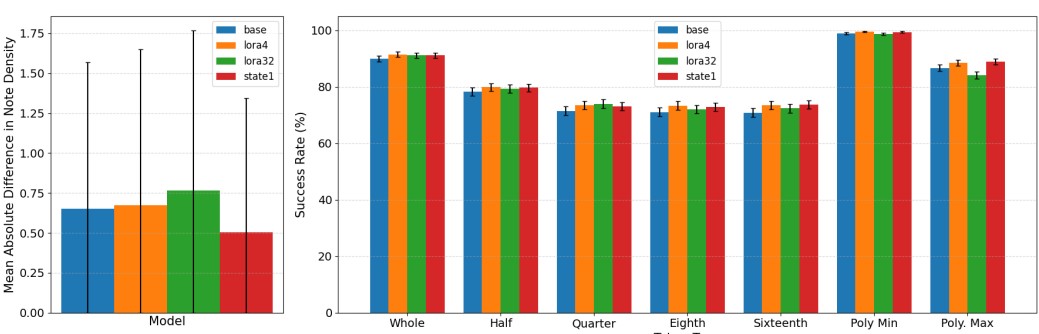

Figure 11: Evaluation of attribute control effectiveness on the 2-bar $N = 2, C = 8$ objective.

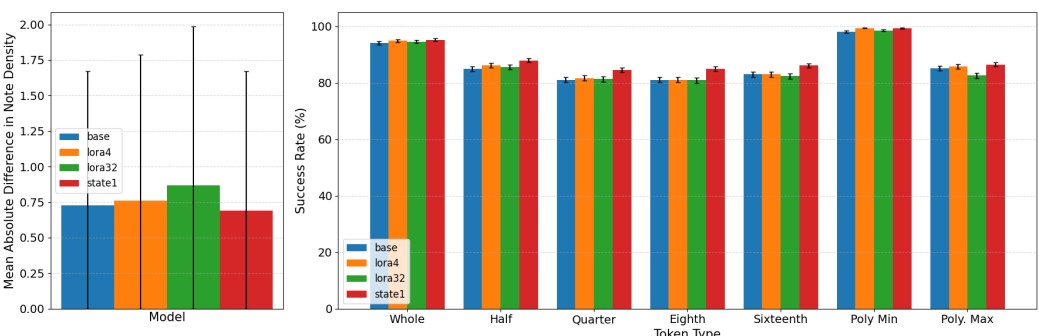

Figure 12: Evaluation of attribute control effectiveness on the 4-bar $N = 4, C = 16$ objective.

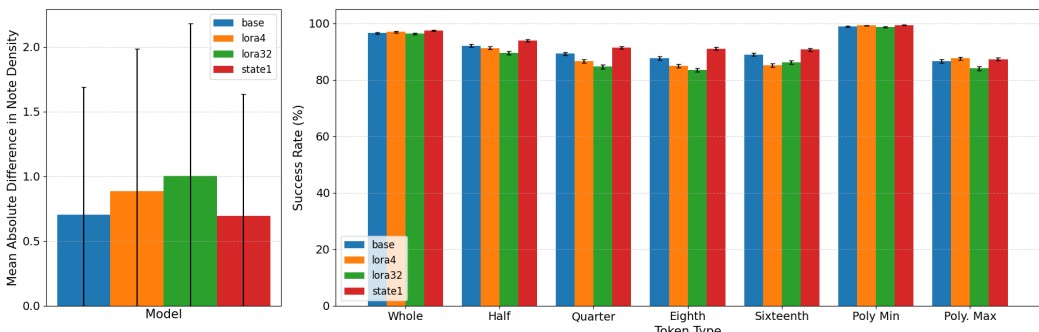

Figure 13: Evaluation of attribute control effectiveness on the 8-bar $N = 8, C = 32$ objective.

Table 16: Random infilling evaluation of MIDI-RWKV vs. CA2 on the GigaMIDI test set.

| Model | CP ↑ | GS ↑ | PCHE ↓ | F1 ↑ |
|---|---|---|---|---|
| 8-bar RWKV | **0.694 ± 0.141** | 0.943 ± 0.022 | 0.300 ± 0.318 | 0.219 ± 0.153 |
| 8-bar CA2 | 0.472 ± 0.303 | **0.959 ± 0.049** | **0.122 ± 0.106** | **0.771 ± 0.832** |
| 16-bar RWKV | **0.682 ± 0.164** | 0.940 ± 0.044 | 0.278 ± 0.268 | 0.342 ± 0.256 |
| 16-bar CA2 | 0.486 ± 0.242 | **0.951 ± 0.082** | **0.107 ± 0.134** | **0.824 ± 0.419** |

Table 17: Single-section infilling evaluation of MIDI-RWKV and Polyffusion on a POP909 test set.

| | CP ↑ | GS ↑ | PCHE ↓ | F1 ↑ |
|---|---|---|---|---|
| 2-bar RWKV | **0.316 ± 0.182** | **0.947 ± 0.039** | **0.497 ± 0.415** | **0.063 ± 0.146** |
| 2-bar Polyffusion | 0.271 ± 0.206 | 0.927 ± 0.017 | 0.505 ± 0.401 | 0.044 ± 0.064 |
| 4-bar RWKV | **0.293 ± 0.159** | **0.925 ± 0.052** | **0.433 ± 0.348** | **0.072 ± 0.107** |
| 4-bar Polyffusion | 0.208 ± 0.190 | 0.898 ± 0.028 | 0.450 ± 0.163 | 0.055 ± 0.021 |
| 8-bar RWKV | **0.287 ± 0.116** | **0.884 ± 0.062** | **0.314 ± 0.276** | **0.054 ± 0.072** |
| 8-bar Polyffusion | 0.163 ± 0.132 | 0.869 ± 0.018 | 0.522 ± 0.159 | 0.049 ± 0.062 |

We observe that MIDI-RWKV performs moderately better on all metrics than Polyffusion. Given the different modalities (diffusion vs. autoregression), two-year distance between releases, and vast differences in size and quality of the training data, we do not believe there is enough signal to draw any conclusions from this data, other than that *it is possible for* high-quality multi-track foundation models to outperform specialized models at downstream tasks (e.g. single-track infilling, as here)—which has been known in other modalities, like text, for a long time already. Diffusion remains a very natural modality for infilling and we hope future work will explore it further.

### E.6 Example Generations

The following are piano roll visualizations of some infillings of POP909 songs, with the original song on the left and the generated content on the right. The infilled section is highlighted in yellow on both sides, although no change was made to the original song; there is also no change to the non-highlighted section on either side, which is provided as context. The MELODY track is represented in blue, the BRIDGE track in orange, and the PIANO track in green. Note that the context following the infilled section changes slightly between the original and generated content, due to changes of notes that began in the infilled section and were held into the following context.

Unlike in our subjective evaluation, all three tracks were infilled. Each track was infilled individually over a contiguous section of 4 (Figure 14) or 8 (Figure 15) track-measures, representing a fairly long-context task (12 or 24 infilled track-measures in total, and three times that in total context). We omit shorter-context tasks for the sake of space, and because they typically look very similar to the ground truth, but we believe these long-context generations represent a lower bound on the performance of the model. We invite the reader to use the demo notebook provided in the supplementary material to test further generations.

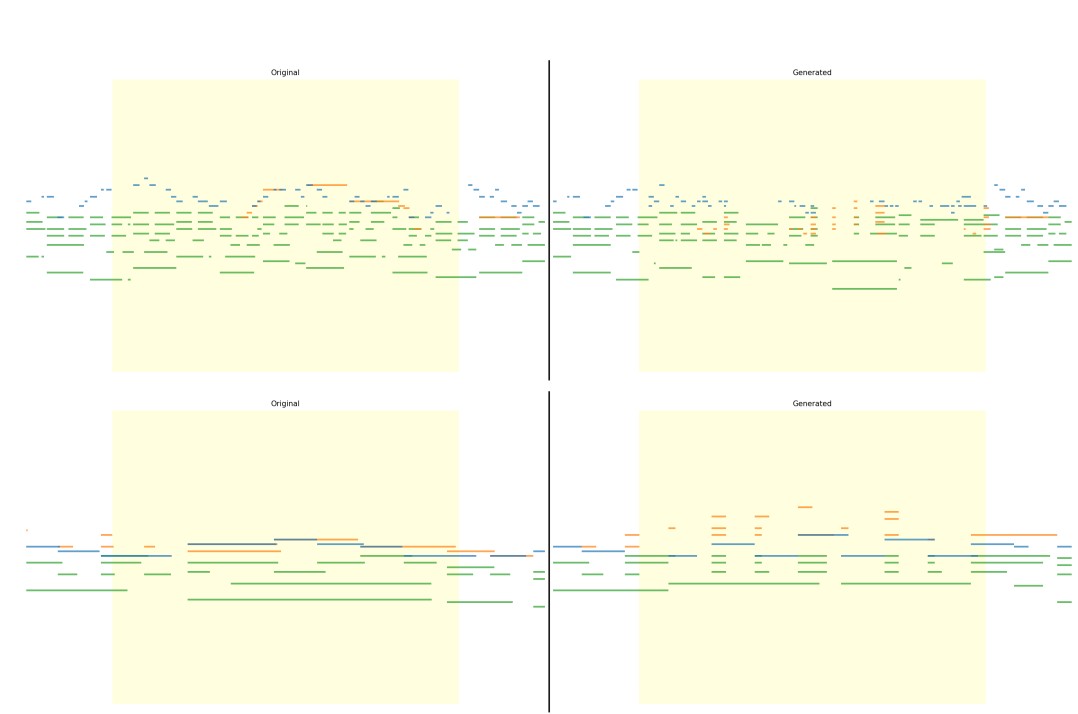

Figure 14: Two example generations with $N = 4, C = 16$ per track.

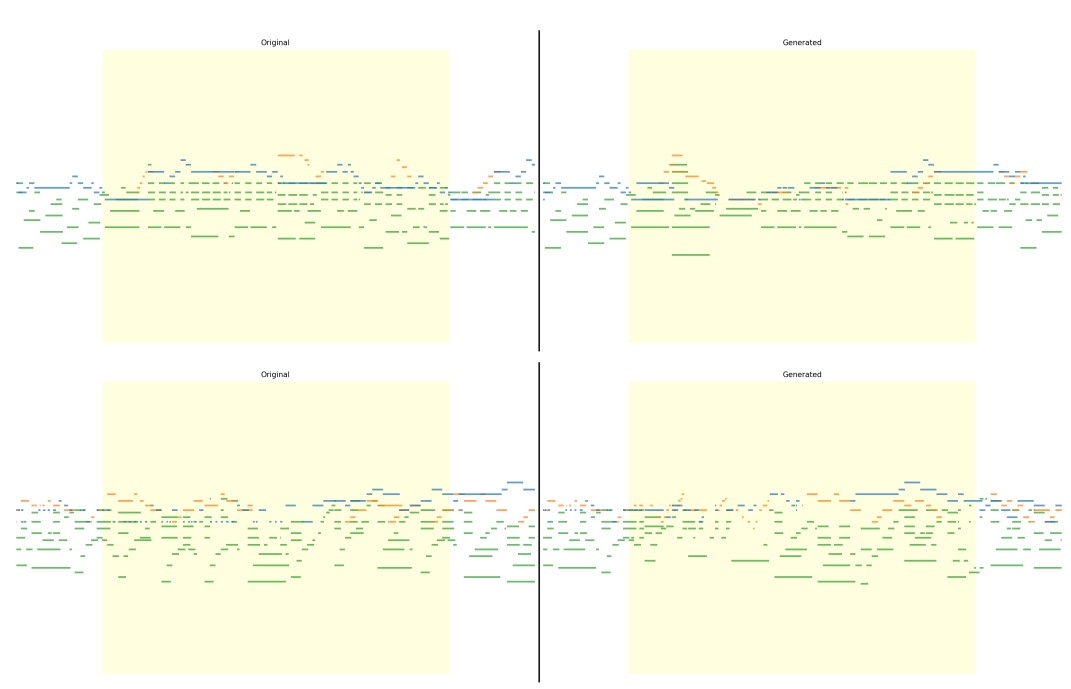

Figure 15: Two example generations with $N = 8, C = 32$ per track.

