# OpenReview forum: "Adaptable Symbolic Music Infilling with MIDI-RWKV"
_ICLR.cc/2026/Conference — Submitted to ICLR 2026_

### Official Review · Reviewer_1Rax · 2025-10-31

**Soundness:** 2
**Presentation:** 3
**Contribution:** 2
**Rating:** 2
**Confidence:** 4

**Summary:**

The paper introduces MIDI-RWKV, a 35M-parameter symbolic music infilling model based on the RWKV-7 linear attention architecture, trained on the GigaMIDI dataset for multi-track, long-context, and controllable music generation. It proposes "state tuning" as an efficient method for low-sample style adaptation by optimizing the model's initial hidden states. The authors evaluate the model on objective metrics (e.g., content preservation, groove similarity) and a subjective listening test, claiming it outperforms baselines like MIDI-GPT and Composer's Assistant in infilling tasks.

**Strengths:**

The paper is clearly written, with well-organized sections, informative figures (e.g., comparisons of encoding schemes and infilling representations), and a logical flow from motivation to experiments.

It contextualizes the work adequately within related literature on symbolic music generation and linear architectures. In terms of quality, the experimental setup leverages established datasets (GigaMIDI and POP909) and metrics, and the release of code, models, and supplementary materials enhances reproducibility and potential for community use.

The application of RWKV-7 to symbolic music infilling addresses practical limitations like long-context handling on edge devices, which could be significant for computer-assisted composition workflows. State tuning represents a modest originality in adapting linear models for low-data style transfer, building on prior ideas in RNN initialization but applying them to a music domain.

**Weaknesses:**

The core innovation is limited: the model primarily involves training an RWKV-7 backbone on GigaMIDI with existing REMI+ encoding and Bar-Fill infilling objectives (adapted from Pasquier et al., 2025), followed by fine-tuning on POP909. This lacks significant new scientific insights, as the infilling techniques and attribute controls are derived from prior works (e.g., von Rütte et al., 2023; Huang & Yang, 2020), and state tuning, while efficient, is an extension of established RNN initialization concepts (e.g., Gers et al., 2002) without deep theoretical novelty for ML.

The baselines are insufficiently comprehensive; comparisons focus on models like MIDI-GPT, MMM, and Composer's Assistant, which are not the most mainstream or recent in symbolic music generation. Stronger baselines such as Text2MIDI (Rizzotti et al., 2025, which supports infilling via inference alignment), Text2MIDI-InferAlign, or the Multi-Track Music Transformer (Yu et al., 2022) could provide better context, especially for long-range dependencies and controllability.

The subjective listening test is mentioned but lacks methodological details in both the main text and appendix (e.g., number of participants, their musical training, evaluation criteria like coherence or preference scales, inter-rater reliability), making it hard to assess its validity—adding these would strengthen claims of practical utility.

Overall, the work offers few reusable ML insights (e.g., beyond music-specific applications) and seems better suited for music-focused venues like ISMIR rather than ICLR, where broader algorithmic or architectural advancements are emphasized.

**Questions:**

1. Could you elaborate on the subjective listening test methodology? For instance, how many participants were involved, what was their level of musical expertise (e.g., trained musicians vs. general listeners), what specific criteria were used for rating (e.g., musical coherence, style fidelity, overall preference), and were there measures for inter-rater agreement? Providing these details could bolster the qualitative claims.

2. Why were baselines limited to models like MIDI-GPT and Composer's Assistant? Comparing with more recent or versatile systems such as Text2MIDI (which can handle infilling through inference techniques), Text2MIDI-InferAlign, or the Multi-Track Music Transformer could better demonstrate MIDI-RWKV's advantages in long-context or multi-track scenarios. If these were considered but excluded, what were the reasons?

3. The paper emphasizes state tuning's efficiency for low-data adaptation, but how does it generalize beyond POP909 melodies? For example, have you tested it on other styles or datasets, and what theoretical justifications support its superiority over LoRA in RWKV specifically? Ablations on varying sample sizes (e.g., 10 vs. 99) could clarify its robustness.

4. The contributions seem tailored to music generation; what broader reusable insights does this offer the ICLR community, such as in other sequence modeling domains (e.g., text or code infilling with linear architectures)?

**Details Of Ethics Concerns:**

The paper proposes a subjective evaluation, which should be conducted under approval from ethics committee.

---

> ### Author Response · Authors · 2025-11-15
> **Response by Authors (1/2)**
>
> We thank you for your detailed feedback and hope to offer resolutions to your concerns below! We have uploaded a new revision of the paper that should address many of your questions and concerns. Our comment is split into two to facilitate a comprehensive response within the character limit.
>
> **>  Ethics concerns.** To address your ethics flag first, we performed our subjective evaluation with the approval of the appropriate IRB, as mentioned in the ethics statement. Naturally, we cannot release the approval number due to anonymity constraints. We wanted to release this rebuttal as soon as possible to address this concern.
>
> **> Could you elaborate on the subjective listening test methodology?** Indeed, and it was an oversight on our part for not doing so in Appendix B after spending such effort documenting everything else! We will answer your specific questions here and have included a more comprehensive description in Appendix B.4 of the latest revision. We did mention in Section 5.2 that there were 28 participants, but this number has also been included in Section 4.3 in the latest revision for clarity.
>
> Participants were asked to describe their musical training as “Amateur”, “Intermediate”, and “Expert” (elaborated on in the revised paper - roughly “little to no formal schooling”, “some formal schooling”, and “work in industry/have degree”). The split of respondents was 7/15/6 respectively, though we expect several applicants were hesitant to self-describe as “Expert”s. They were shown four clips at a time in randomized order, one from each model tested, and were able to replay each at will. They were then asked to rank the clips in order of overall preference; we eschew numerical ratings at the advice of work such as Yannakakis and Martinez [1].
>
> We also have added some new statistical results to Appendix E.2 and may move them to the main text contingent on space restrictions. We augmented our Wilcoxon signed-rank test with a Friedman test on the collective responses, to demonstrate that the models produce different quality outputs as assessed overall by the respondents, and performed a per-question analysis of Kendall’s W to determine inter-rater agreement. The Friedman test is significant with $p < 0.001$, and six of the ten questions are significant for Kendall’s W with $p < 0.05$, with $W = 0.20$ to $0.25$.
>
> **> Why did baselines not include more recent work?** We intentionally limit our comparisons to models in the “family” of DAW-integrated infilling models: MMM and MIDI-GPT are integrated with Calliope, and CA/CA2 are integrated with Reaper. Indeed, MIDI-RWKV is a member of this family; we are working on integrating it with Calliope as we speak. Such models are designed to be tightly integrated with composers’ workflows - see for instance Tchmeube et al. [2] - to such an extent that they feel natural to use, enabling true human-AI cocreation.
>
> In particular, this constrains the model to be sufficiently lightweight to run on composers’ existing hardware, which frequently lack dedicated GPUs. CA2, at almost 200M parameters, pushes the bounds of what is feasible; larger foundation models like Text2MIDI or MuseCoco, at 1B+ parameters, are simply too large and slow for most composers to use locally. The objective is not to compete with “mainstream” models, as you may suggest; these models generally aim to do full-song generation from scratch or continuation alone, tasks less relevant to an artist seeking to refine a composition. This note has been added to Section 5.1.
>
> Therefore, we believe it does not matter how such models compare to MIDI-RWKV because they target fundamentally different niches. It is much like comparing a plane to a car: the plane can travel across a country (in the analogy, generate a full song from scratch) much faster, but one would not use it to do more frequent, lightweight tasks like their daily groceries (edit a section of an existing song). Likewise, a road trip is less practical for crossing a country than a flight. Both are important!
>
> Additionally, most such systems (such as Text2MIDI) are unidirectional in nature, only looking at past context, which prevents them from being adapted to infilling tasks that require both past and future context and thus prevents direct “apples-to-apples” comparison on infilling tasks. We are not aware of the “inference techniques” that you mention to adapt Text2MIDI for infilling. Could you please elaborate on those?
>
> **Our response is continued in the following comment.**
>
> [1] Georgios N. Yannakakis and Hector P. Martinez. Ratings are overrated! Frontiers in ICT Vol. 2, 2015.
>
> [2] Renaud Bougueng Tchemeube et al. Evaluating human-ai interaction via usability, user experience and acceptance measures for MMM-C: a creative AI system for music composition. IJCAI 2023.

---

> ### Author Response · Authors · 2025-11-15
> **Response by Authors (2/2)**
>
> **This response is continued from the previous comment.**
>
> **> Does state tuning generalize beyond POP909?** We performed initial tests on the full corpus of the Beatles and saw similarly impressive results, but decided to publish POP909 results due to copyright concerns. We will perform some ablations on much smaller sets (on the order of ~10 training examples) over the coming weeks, so please stay tuned for those in future revisions; we wished to begin the discussion while those are ongoing, but please refer to the new Appendix D.2 for some more in-depth empirical analyses of state tuning.
>
> Regarding theoretical justifications, we refer you to the second paragraph of Section 3.4 for our justification of why state tuning should work, which is newly supported by the evidence in Appendix D.2. In brief, we believe it should work better than LoRA because in the state space of a recurrent model like RWKV, style adaptation is best formulated as an initial-value problem, as the latent space already contains priors about the style from the training data - that is, as long as the style is in distribution, which is less and less of a problem with the rapid scaling of data in music pretraining corpuses. Thus, the model should not require low-rank adapters to add additional information about that style.
>
> **> The contributions seem tailored to music generation.** Perhaps regrettably, *we agree with you to a large extent!* And you are completely correct in that many of our techniques are derived from prior works. Despite that, we hope to convince you that our modern state tuning results - bringing a technique studied on classical RNNs into the Transformer era and adapting it to contemporary tasks - are not unworthy of ICLR, and neither are music-specific advancements any less worthy than many of the text-specific advancements that will inevitably pass the review process. We accept that the following argument may not be convincing to you and, regardless of your ultimate decision on this point, we thank you for your effort to provide insightful feedback on our work. The same concern was raised by Reviewers 4eqD and mHKN, so please forgive our reuse of much of the same argument.
>
> We dedicated a fairly large portion of the paper to state tuning, which we emphasize is not unique to music modeling or even RWKV alone: it is applicable to any recurrent or hybrid-recurrent model, and as recurrent models gain traction in other fields like text modeling (see for instance Qwen3-Next, MiniMax-M1, et c.) it is useful to explore what alternative methods these models admit beyond being asymptotically faster at training and inference.
>
> Our objective and subjective results plus the analysis in Appendix D.2 provide, to our knowledge, the first empirically-supported analysis of state tuning applied to a pretrained post-Transformer recurrent model. We specify post-Transformer as these models (RWKV, Gated DeltaNet, and so on) have had the benefit of learning from years of work on Transformers, and so their design and in particular dynamics will differ greatly from the recurrent models of the 2010s and before. This opens the door for a new, lightweight method to adapt pretrained recurrent models in other fields, such as text. Perhaps the paper would benefit from reframing around this contribution?

---

> > ### Comment · Reviewer_1Rax · 2025-11-15
> >
> > Thanks. I looked through your rebuttal and I like it. Although my concern about the contribution impact still stands, I raised the score from 2 to 4, and I keep the oppotunity to raise it again at the end of rebuttal.

---

> > > ### Author Response · Authors · 2025-11-19
> > >
> > > Thank you for your response! We will do our best to address the contribution impact with further empirical results on state tuning. We have already added some more results in Appendix D.2 on the trajectory during training, showing that state tuning is able to rapidly find local minima, and we should have extra results on further datasets in the next week or so.
> > >
> > > Please do not hesitate to let us know if there is anything in particular you think would improve the paper, and we will update you when relevant additions have been made to the revision!

---

> > > > ### Author Response · Authors · 2025-11-27
> > > > **Update by Authors**
> > > >
> > > > We would like to let you know that the results we mentioned in our previous comment have now been added to Appendix D.4, on four new finetuning datasets: Pop1K7 (1747 pop pieces), YM2413-MDB (669 videogame soundtracks), the Nottingham dataset (1037 folk songs), and the “waltzes” subset of the Nottingham dataset (35 waltzes). These were also not limited to melody infilling like our POP909 results were. The results are in Tables 10-13 in the appendix. We observe that across various styles and datasets - even in the heavily sample-constrained regime Nottingham “waltzes” subset, training on only 4 songs - state tuning outperforms LoRA finetuning, and we hope these results address more of your concerns about the finetuning evaluation!

---

### Official Review · Reviewer_4eqD · 2025-10-31

**Soundness:** 3
**Presentation:** 3
**Contribution:** 2
**Rating:** 4
**Confidence:** 3

**Summary:**

This paper introduces MIDI-RWKV, a foundation model for symbolic music infilling designed for computer-assisted composition. The model is based on the RWKV-7 architecture , a linear-time architecture that allows MIDI-RWKV to be relatively small (35M parameters ) and efficient, making it suitable for edge devices and long-context applications.
The system aims to be controllable, using attribute controls for features like note density and polyphony, and adaptable. For adaptability, the paper's main methodological contribution is the exploration of state tuning—finetuning the model's initial hidden state rather than its weights—for low-sample style adaptation.
The authors conduct a thorough evaluation:
- They compare the base MIDI-RWKV against other infilling models like Composer's Assistant and MIDI-GPT on several objective metrics (e.g., Content Preservation, Groove Similarity) , showing that their model is competitive or superior, particularly given its smaller size.
- They directly compare state tuning against LoRA finetuning on a style adaptation task using the POP909 dataset.
- The results, supported by both objective metrics and a subjective listening test , demonstrate that state tuning provides superior adaptation in a low-sample regime, being preferred by participants over both the base model and LoRA-tuned variants.

**Strengths:**

High Quality and Empirical Thoroughness: The paper's greatest strength is its high-quality, thorough, and well-conducted empirical evaluation. The authors compare their 35M parameter model against several relevant baselines, including a larger one (CA, 54M), and demonstrate its effectiveness. The validation of the state tuning method is particularly strong, as it includes objective metrics, a subjective listening test, and stability analysis.

Practical Significance: The paper tackles a significant and practical problem for musicians: creating controllable generative tools that are efficient enough to run on local, "edge" devices. By using the efficient RWKV-7 architecture and developing a small 35M parameter model, this work is a positive step towards democratizing such tools.

Novel Application of a Method: While "state tuning" is not a new concept (as the paper admits 34), its application to style adaptation in a modern, large-scale generative model is a key strength. The paper provides a strong demonstration that this
parameter-efficient method (training only $L \times d$ parameters) can be more effective than LoRA in a low-sample regime, which is a valuable finding for the community.

**Weaknesses:**

$\bullet$ Limited Conceptual Novelty: As detailed in the "Contribution" section, the paper's primary weakness is its lack of fundamental research novelty. The work is a clever and effective combination of existing components (RWKV-7 architecture , REMI+ encoding , Bar-Fill objective , and state tuning ). This makes the paper feel more like a strong technical report or an application paper rather than a new research contribution for a conference on learning representations.

$\bullet$ Limited Analysis of State Tuning: The paper shows that state tuning works well, but it doesn't deeply investigate why. The rationale in Section 3.4 is a hypothesis ("directs the hidden state evolution to operate within a new subspace" ). The paper would be significantly strengthened if it included experiments to verify this, for instance, by analyzing and visualizing the hidden states of the base vs. tuned models to show this "subspace" shift. Without this, the contribution remains a "black box" empirical finding.


$\bullet$ Infilling Objective Limitations: The "single-section infilling" objective  is a significant simplification of the arbitrary infilling problem. The authors rightly note this avoids the "exponentially" large space of masking patterns. However, this comes at the cost of inference-time latency. To emulate the arbitrary infilling (bottom of Fig 3), the system must perform "repeated single-section infilling". The authors concede this "currently takes somewhat longer than with Composer's Assistant". This serial, repetitive inference process could potentially negate the training and per-token efficiency gains of the RWKV architecture, especially for complex infilling requests from a user. This trade-off is not quantified.

**Questions:**

$\bullet$ On the Novelty of State Tuning: The paper correctly cites prior work on training the initial state of RNNs. Could you please clarify precisely what the novel contribution of your "state tuning" method is, beyond applying this existing concept to the RWKV architecture for style adaptation? Is there a methodological difference in how you optimize the state, or is the contribution purely the empirical demonstration of its effectiveness in this new context?

$\bullet$ On Inference Latency: The "single-section infilling" approach requires "repeated single-section infilling"  to handle arbitrary masking patterns. This seems to introduce a significant serial dependency at inference time. Could you quantify this latency? For example, for the "arbitrary masking pattern" shown in Figure 3 (bottom), how many sequential model calls would be required, and how does the total wall-clock time compare to a single call from a model that can handle arbitrary masking (like Composer's Assistant)?

---

> ### Author Response · Authors · 2025-11-15
> **Response by Authors (1/2)**
>
> Thank you for your in-depth feedback, and we particularly appreciate that you provided suggestions for experiments as well! We are pleased to inform you that we were able to run some extra experiments, which have been added to the latest revision of the paper, and we intend to run some more over the course of the rebuttal period. In particular, we hope that the increased analysis of state tuning will satisfy you, although we welcome further suggestions. Our comment is split into two to facilitate a comprehensive response within the character limit.
>
> **> On the limited analysis of state tuning.** We have run some quick preliminary experiments to visualize the effects of state tuning. We tested on all three models of our stability tests to incorporate diversity among training runs and increase the sample size. These results include a PCA visualization of overall model hidden states as well as individual prompt trajectories, quantification of the distance between the state-tuned state and untuned state (so as to demonstrate that the trajectories remain disjoint in state space), and signal-to-noise ratio for these results.
>
> These results currently reside in the new Appendix D.2 of the latest revision while we figure out where they will be best located. We think the PCA analysis in particular is rather conclusive as far as the claim is concerned, but we hope to reinforce it further with more experimental results.
>
> We intend to run further experiments on how these results change as a function of finetuning steps ([Edit 11/18/2025] these have been added in the latest revision), and add more results as we perform other state tuning experiments for Reviewers 1Rax and mHKN. If you believe the claims would be further strengthened by adding any other visualizations or results, please do not hesitate to let us know, as your feedback on this matter is warmly welcomed!
>
> **> On inference latency.** We indeed did not quantify the time trade-off of single-section infilling in the paper, which was an oversight. We have added histograms for the time taken for 100 inference requests for 8- and 16-bar random infilling (the objective of CA) of both MIDI-RWKV and CA to the new Appendix E.1. These were run with batch size 1 and a single i9-14900K. To go from less than 1 second on average per 8-bar request (CA) to just under 5 seconds on average (ours) is a drastic slowdown, but we believe that such a slowdown is not significantly important in practice, as it remains a negligible amount of time; see similar waiting times proving a non-issue in Tchmeube et al. [1]. We may move these results to the main text contingent on space restrictions.
>
> In any case, the trade-off will differ on a case-by-case basis, and will largely depend on the infilling format: the more disjoint track-sections that must be infilled, the greater the difference, due to the exacerbation of the sequential dependency you rightfully point out. In Figure 3 (bottom), three calls would be required, due to there being three disjoint track-sections (sections of a track); but in practice this can be made significantly more efficient by implementing inference tricks like state reuse we mentioned earlier. As we continue our integration with Calliope and move towards composer integrations, we will likely turn more attention to such matters, but as it stands our method is still regrettably slower.
>
> **> On the novelty of state tuning.** Our novel contribution is in demonstrating objective and subjective improvements in state tuning using direct backpropagation on the hidden state, and with the addition of your recommended experiments in Appendix D.2, the first empirical analyses (to our knowledge) of state tuning and justification for the “subspace shift” claim you rightfully point out.
>
> Mohajerin and Waslander [2], whom we cite in the text, train a FFN on truncated context to initialize a larger RNN and do not explore interpretability of their methods, and more recent work by Xiao et al. [3] also train a smaller model to replicate RNN dynamics and provide minimal explanations or analysis of their results. In contrast, our work (now) provides quantitative and qualitative analyses that demonstrate how state tuning works mechanistically and where in the representational space these improvements manifest.
>
> **Our response is continued in the following comment.**
>
> [1] Renaud Bougueng Tchemeube et al. Evaluating human-ai interaction via usability, user experience and acceptance measures for MMM-C: a creative AI system for music composition. IJCAI 2023.
>
> [2] Nima Mohajerin and Steven L. Waslander. State initialization for recurrent neural network modeling of time-series data. IJCNN 2017.
>
> [3] Liu Xiao et al. State tuning: state-based test-time scaling on RWKV-7. arXiv:2504.05097.

---

> ### Author Response · Authors · 2025-11-15
> **Response by Authors (2/2)**
>
> **This response is continued from the previous comment.**
>
> **> On overall limited conceptual novelty.** The same concern was raised by Reviewers 1Rax and mHKN, so please forgive our reuse of much of the same argument. Perhaps regrettably, we agree with this point to a large extent! And many of our techniques are indeed derived from prior works. Despite that, we hope that our modern state tuning results - bringing a technique studied on classical RNNs into the Transformer era and adapting it to contemporary tasks - are not unworthy of ICLR, and neither are music-specific advancements any less worthy than many of the text-specific advancements that will inevitably pass the review process. We accept that the following argument may not be convincing to you and, regardless of your ultimate decision on this point, we thank you for your effort to provide insightful feedback on our work.
>
> We dedicated a fairly large portion of the paper to state tuning - even more at your expert advice - which we emphasize is not unique to music modeling or even RWKV alone: it is applicable to any recurrent or hybrid-recurrent model, and as recurrent models gain traction in other fields like text modeling (see for instance Qwen3-Next, MiniMax-M1, et c.) it is useful to explore what alternative methods these models admit beyond being asymptotically faster at training and inference.
>
> Our objective and subjective results plus the analysis in Appendix D.2 provide, to our knowledge, the first empirically-supported analysis of state tuning applied to a pretrained post-Transformer recurrent model. We specify post-Transformer as these models (RWKV, Gated DeltaNet, and so on) have had the benefit of learning from years of work on Transformers, and so their design and in particular dynamics will differ greatly from the recurrent models of the 2010s and before. This opens the door for a new, lightweight method to adapt pretrained recurrent models in other fields, such as text. We hope this constitutes sufficient novelty for a venue such as ICLR, although the paper may need some reframing, and we understand that you may disagree. Your feedback on this, as with all else, will be much appreciated.

---

### Official Review · Reviewer_mHKN · 2025-11-01

**Soundness:** 3
**Presentation:** 4
**Contribution:** 2
**Rating:** 2
**Confidence:** 4

**Summary:**

This paper proposes a new training pipeline for symbolic music generation using RWKV and state tuning.

**Strengths:**

1. The application of RWKV to MIDI LM is an important work for the music AI community.
2. The results on state tuning vs LoRA is pretty interesting.
3. Nice narrative and writing.

**Weaknesses:**

1. The major limitation is the novelty. The encoding scheme, model architectures are all well-defined, making it a good application paper but less ideal for a ICLR paper.
2. The setting of single-section infilling is limited compared to abitrary masking.
3. Missing comparison against some types of symbolic infilling models, like diffusion-based [1].
4. The fine-tuning datasaet is limited to only POP909. If only 99 songs are needed for training there are many other genres can be experimented on, including folk songs (Nottingham, BFDB), classical piano, pop piano (ailabs1k7), pop (rwc pop) etc. Should be  easy to at least get some demos and objective evaluation on them.
5. In table 2 & 3, it would be better to add the evaluation metrics for the ground-truth (human).
6. The subjective evaluation is not performed against existing baselines.

[1] Min, L., Jiang, J., Xia, G., & Zhao, J. (2023). Polyffusion: A diffusion model for polyphonic score generation with internal and external controls. arXiv preprint arXiv:2307.10304.

**Questions:**

1. Line 098: A real-world efficiency comparison between MIDI RWKV and transformer-based model would be cool.
2. Line 228: Why "the analog would be the KV cache"? It looks more similar to (trainable) prefix tokens to me.

---

> ### Author Response · Authors · 2025-11-15
> **Response by Authors**
>
> Thank you for your feedback and we hope to address your concerns below. We have uploaded a revised paper and supplemental to address many of your concerns.
>
> **> Real-world efficiency comparison.** We have added this in the latest version to the new Appendix E.1; on a 8-bar random task the average latency increases from less than 1 second (CA) to just under 5 seconds (ours). We believe that such a slowdown is not significantly important in practice, as it remains a negligible amount of time, given the work produced by the model for its user; see for instance similar waiting times proving a non-issue in Tchmeube et al. [1]. We may move these results to the main text contingent on space restrictions.
>
> **> Single-section infilling is limited.** We can model any arbitrary masking input with single-section infilling, it just takes a bit longer currently (see above); please see our discussion in Section 3.3.
>
> **> Missing comparison.** We did not compare against Polyffusion due to constraints on its output schema, most notably that its binary 2-channel representation is inadequate for arbitrary-track infilling, which is a key tenet of our model and those we compare against (see Table 1). Therefore it would be impossible to do a fair comparison, as Polyffusion would fail on the vast majority of our test samples. We nonetheless believe that Polyffusion is an important model and have added it to our related works section in the latest revision!
>
> **> Limited fine-tuning experiments.** We have not yet had time to do experiments on this, as we wished to begin the discussion while these are ongoing, but we will commit to doing further finetuning experiments over the course of the rebuttal period and update you as we get results. Thank you in particular for suggesting specific datasets to try!
>
> **> Tables 2 & 3.** Could you please elaborate on what you mean? Since the metrics are calculated with respect to the ground truth, the ground truth would get a perfect 100% on all metrics, unless we misinterpret your suggestion.
>
> **> Limited subjective evaluation.** We are currently conducting a parallel subjective evaluation of the model against existing baselines, but it will unfortunately not be completed in time for rebuttals. We were unable to do it previously because we wished to use the initial study to emphasize the efficacy of the state tuning method, and asking users to review more than 4 audio snippets at a time would have decreased the quality of the results.
>
> **> Analog of state vector.** The KV cache analogy refers to the functional role. In Transformers, the KV cache stores past context to avoid recomputation during generation. Likewise, the RNN state encapsulates all prior context in a fixed-size representation that gets updated incrementally. You are correct in that it is also analogous to prefix tokens in terms of carrying contextual information, but the distinction is that the state evolves over the course of training and inference (hence why state tuning works) whereas prefix tokens are fixed.
>
> **> Limitation in novelty.** The same concern was raised by Reviewers 4eqD and 1Rax, so please forgive our reuse of much of the same argument. *We agree with this point to a large extent!* But we hope that our modern state tuning results - bringing a technique studied on classical RNNs into the Transformer era and adapting it to contemporary tasks - are not unworthy of ICLR, and hope the following argument may convince you of this. Regardless of your ultimate decision, we thank you for your effort to provide detailed feedback on our work.
>
> We dedicated a fairly large portion of the paper to state tuning, which we emphasize is not unique to music modeling or even RWKV alone: it is applicable to any recurrent or hybrid-recurrent model, and as recurrent models gain traction in other fields like text modeling (see for instance Qwen3-Next, MiniMax-M1, et c.) it is useful to explore what alternative methods these models admit beyond being asymptotically faster at training and inference.
> Our objective and subjective results plus the analysis in Appendix D.2 provide, to our knowledge, the first empirically-supported analysis of state tuning applied to a pretrained post-Transformer recurrent model. We specify post-Transformer as these models (RWKV, Gated DeltaNet, and so on) have had the benefit of learning from years of work on Transformers, and so their design and in particular dynamics will differ greatly from the recurrent models of the 2010s and before. This opens the door for a new, lightweight method to adapt pretrained recurrent models in other fields, such as text. We hope this justifies the venue ICLR, although the paper may need reframing, and we understand that you may disagree. Your continued feedback will be much appreciated.
>
> [1] Renaud Bougueng Tchemeube et al. Evaluating human-ai interaction via usability, user experience and acceptance measures for MMM-C: a creative AI system for music composition. IJCAI 2023.

---

> > ### Comment · Reviewer_mHKN · 2025-11-15
> >
> > Thank you for your clarifications!
> > They address several of my previous concerns (regarding single-section infilling and Tables 2 and 3).
> >
> > Together with other reviewers’ comments and your responses, one main issue is that it is very difficult to have a clear understanding of the model’s performance for several reasons:
> >
> > **Baselines**
> >
> > Baseline CA2:
> > The authors should report the results regardless of model size differences. It is acceptable if the proposed model performs worse; such results are still valuable for understanding the model’s behavior and limitations.
> >
> > Diffusion baseline:
> > Ideally, the authors could train a diffusion-based model of comparable size to ensure a fair comparison. However, if time constraints make this infeasible, they could at least compare against existing diffusion-based models (e.g., Polyffusion) on a task that both models claim to support.
> > For example, if Polyffusion does not support multi-track inpainting, the comparison can be made on single-track inpainting instead. Comparisons should not be limited to tasks uniquely supported by the proposed model.
> >
> > **Subjective Evaluation**
> >
> > No subjective evaluation results are currently provided.
> >
> > **Demo and Case Study**
> >
> > No case study or demo has been presented. Please add some so the reader can make sense of the model's performance.

---

> > > ### Author Response · Authors · 2025-11-19
> > >
> > > We appreciate the identification of this weakness and have worked to address them. Please let us know what lingering concerns may remain after this!
> > >
> > > **Regarding baselines**, you are right that reporting CA2 baselines would help contextualize the model’s performance, and really there is no reason not to report them other than to save space in the main text; we have thus added these results to Appendix E.5 and attached them below. The proposed model performs worse on all metrics except content preservation, surprisingly. We suggest this may be due to CA2’s default sampling parameters tending towards reproducing the ground truth context verbatim, hence the high F1 scores.
> > >
> > > | Model | CP ↑ | GS ↑ | PCHE ↓ | F1 ↑ |
> > > |---|---|---|---|---|
> > > | 8-bar RWKV | **0.694 ± 0.141** | 0.943 ± 0.022 | 0.300 ± 0.318 | 0.219 ± 0.153 |
> > > | 8-bar CA2 | 0.472 ± 0.303 | **0.959 ± 0.049** | **0.122 ± 0.106** | **0.771 ± 0.832** |
> > > | 16-bar RWKV | **0.682 ± 0.164** | 0.940 ± 0.044 | 0.278 ± 0.268 | 0.342 ± 0.256 |
> > > | 16-bar CA2 | 0.486 ± 0.242 | **0.951 ± 0.082** | **0.107 ± 0.134** | **0.824 ± 0.419** |
> > >
> > > We also do not understand your focus on diffusion models, nor the strange framing to particularly focus on Polyffusion, seeing as none of the proposed contributions relate to the autoregression/diffusion divide. Regardless, we have gathered some results for single-track infilling with Polyffusion, as you suggest. These are below and have been added to Appendix E.5.
> > >
> > > We observe a moderate increase across the board in favor of the proposed model, suggesting MIDI-RWKV generally performs better on this downstream task; however, we cannot draw any particularly significant conclusions about the *methodology* of either study, in light of the age of Polyffusion and difference in amount and quality of training data. We believe diffusion is a promising approach for infilling models, but at the moment our focus is on autoregression.
> > >
> > > | | CP ↑ | GS ↑ | PCHE ↓ | F1 ↑ |
> > > |---|---|---|---|---|
> > > | 2-bar RWKV | **0.316±0.182** | **0.947±0.039** | **0.497±0.415** | **0.063±0.146** |
> > > | 2-bar Polyffusion | 0.271±0.206 | 0.927±0.017 | 0.505±0.401 | 0.044±0.064 |
> > > | 4-bar RWKV | **0.293±0.159** | **0.925±0.052** | **0.433±0.348** | **0.072±0.107** |
> > > | 4-bar Polyffusion | 0.208±0.190 | 0.898±0.028 | 0.450±0.163 | 0.055±0.021 |
> > > | 8-bar RWKV | **0.287±0.116** | **0.884±0.062** | **0.314±0.276** | **0.054±0.072** |
> > > | 8-bar Polyffusion | 0.163±0.132 | 0.869±0.018 | 0.522±0.159 | 0.049±0.062 |
> > >
> > > **Regarding subjective evaluation and demo**, we are unfortunately unable to provide a subjective evaluation comparing multiple models due to time constraints, and remain limited to the finetuning subjective evaluation. Nonetheless, to help contextualize the model’s performance, we have added a minimal Jupyter notebook demo to the supplementary material, as well as some of the base model samples that were provided in the listening test.
> > >
> > > Please note that the notebook demo is not particularly user-friendly at the moment, as it is meant to be run in VSCode, and so lacks the convenient data input formats of Google Colab. We are not currently confident in our ability to share a Colab while maintaining anonymity, so please forgive us for delaying the sharing of a nicer demo. We hope to figure this out before the rebuttal period ends.

---

> > > > ### Author Response · Authors · 2025-11-27
> > > > **Update by Authors**
> > > >
> > > > As we eagerly await your response, we would like to inform you that we have added results on four new finetuning datasets to Appendix D.4: on Pop1K7 (1747 pop pieces), YM2413-MDB (669 videogame soundtracks), the Nottingham dataset (1037 folk songs), and the “waltzes” subset of the Nottingham dataset (35 waltzes). These were also not limited to melody infilling like our POP909 results were. The results are in Tables 10-13 in the appendix. We observe that across various styles and datasets - even in the heavily sample-constrained regime Nottingham “waltzes” subset, training on only 4 songs - state tuning outperforms LoRA finetuning, and we hope these results address more of your concerns about the finetuning evaluation!

---

> ### Comment · Reviewer_mHKN · 2025-11-28
> **Response to authors**
>
> Thanks to the authors for the detailed response, and I am definitely willing to provide a re-evaluation based on the new information the author provided. However, it would be helpful if the author could provide more information on the supplement material's audio files. There is currently no README associated with the files, and I cannot quite understand it.
>
> 1. Which file is the reference piece?
> 2. Which file is the generated piece?
> 3. xxx_base.mid.mp3 and xxx_original.mp3 have different lengths. Why?
> 4. Where is the start and the end of the generation (in seconds for each mp3 file?)
>
> Thanks.
>
> Also for the demo I do not mean the "runnable source code" but rather some music examples to be added onto the paper. It can just be some sheet music score/piano roll images in the appendix, clearly marking the beginning and the end of the generation part. The baselines' generated samples can also be included to make some comparisons. It gives the reader more context of what the model is capable of. It is okay to do this after the rebuttal, but this is crucial for any music generation papers.

---

> > ### Author Response · Authors · 2025-11-28
> > **Response by Authors**
> >
> > Thank you for your response, and we are sorry that we will not be able to continue the discussion further given the PCs’ recent decision to close reviewer responses. Nevertheless we will ensure to address your questions here, and we have added all of this information to a README in the `samples/` folder.
> >
> > 1, 2. The reference piece was `xxx_original.mp3` and the generated piece was `xxx_base.mid.mp3` (for base model). We have updated the naming in the most recent version of the supplemental to `xxx_reference.mp3` and `xxx_generated.mp3` for clarity.
> >
> > 3: Because the model can only represent a quantized range of tempos (these can be found in the `tokenizer.json` around line 800), the tempo of the generated content does not exactly match the tempo of the reference, resulting in a few seconds of difference in length. This can be seen within the first few seconds if the audio clips are played side-by-side in Audacity or a similar program. Discrepancies of a few seconds due to tempo misalignment also occurred in the subjective evaluations of other studies (e.g. CA) so we did not think much of it.
> >
> > 4: Usually the beginning and end are roughly 1/4 and 3/4 of the way through the file, respectively, but the exact numbers have been added to the README.
> >
> > Thank you also for clarifying what is meant by “demo”, and we agree that some sort of visualization of the generated content is important. We have accordingly added some piano roll images to Appendix E.6, which we hope provide a clearer picture of what the model is capable of, and we invite you to explore those examples. Having updated the supplemental and appendix, we hope that this response satisfies your concerns!

---

### Official Review · Reviewer_XRdF · 2025-11-01

**Soundness:** 2
**Presentation:** 3
**Contribution:** 2
**Rating:** 4
**Confidence:** 3

**Summary:**

The paper proposes MIDI-RWKV, a controllable, adaptable multi-track symbolic music infilling model based on RWKV-7. It trains a ~35M-parameter model on GigaMIDI and introduces state tuning (optimizing the model’s initial hidden state) for low-sample style adaptation. Objective evaluations compare against MIDI-GPT and Composer’s Assistant variants; a 28-participant listening test suggests state tuning beats base/LoRA on POP909 melody infillings. Authors say code and weights are included in the supplementary.

**Strengths:**

- Simple, efficient backbone + controllability: Using RWKV-7 for long-context infilling with numerical/categorical controls is well-motivated and clearly described.
- State tuning is a neat adaptation mechanism, parameter-efficient and conceptually distinct from LoRA; the paper positions it clearly.
- Reasonable benchmarking on single-section and random infilling against CA and MIDI-GPT with transparent metrics.
- Human study present (28 participants) with statistical analysis, albeit limited in scope (see weaknesses).

**Weaknesses:**

- No accessible demo/audio page: For a music generation paper, the submission provides no public audio examples or interactive demo; only a claim that “code and weights [are] in the supplementary.” This makes it hard to independently judge musical quality, control fidelity, and usability.
- Evaluation scope undercuts the headline claim: State-tuning experiments and the listening test are confined to POP909 melody-only finetuning, not multi-track use, limiting evidence for the paper’s core “multi-track controllable infilling” pitch.
- Missing stronger baselines in key settings: While the paper lists CA2 as a modern system, it does not compare against it in results; objective tables include CA/MIDI-GPT but not CA2. This weakens claims about competitiveness.
- Human study design is narrow: The user study (28 participants) truncates audio to 4 bars around the infill and compares only finetuned MIDI-RWKV variants (not cross-model comparisons), limiting what we can conclude about musicality and control in realistic multi-track contexts.
- Latency and workflow gaps acknowledged: The method cannot compose in real time, DAW integration is not yet available, and emulating arbitrary-mask infilling via repeated single-section calls is slower than Composer’s Assistant (partly due to under-optimized RWKV inference). This undermines practical impact for ICLR.

**Questions:**

- Audio/demo: Will you provide a public audio page (e.g., curated examples, ablations for control success vs. coherence) and/or a minimal interactive demo (Colab/Gradio)?
- Multi-track evidence: Can you add multi-track subjective tests (not only melody) and show how attribute controls interact across tracks?
- Baselines: Can you include CA2 (or explain why it’s infeasible) and, for POP909, stronger LoRA configurations under matched training budgets/parameter counts?
- Arbitrary-mask infilling: Do you have one-shot arbitrary-mask results/latency (not multi-call emulation), or can you quantify wall-clock latency vs. CA/MIDI-GPT with caching enabled?
- Real-time/DAW path: Any concrete plan or prototype for streaming/real-time token generation and DAW integration (e.g., a Calliope or VST bridge), with preliminary timings?
- State-tuning limits: Could you test extreme/out-of-distribution styles (the paper itself flags this risk) and provide ablations on state-vector size/training steps?

---

> ### Author Response · Authors · 2025-11-15
> **Response by Authors**
>
> Thank you for your feedback! We address your comments below. We have uploaded a revised paper and supplemental to match.
>
> **> No accessible demo/audio page**. We provided instructions to run the inference pipeline in the supplementary README; we would have liked to make a “mini-DAW” piano roll input in Gradio but Gradio sadly was not flexible enough for it. [Edit 11/18/2025] We have added a notebook demo to the supplemental, although it is not terrifically user-friendly; we hope to publish a better Colab demo in the near future.
>
> We did not add audio samples as part of the supplemental because the WAV files were too large to fit and we are not confident in posting them publicly while maintaining anonymity. At your request, however, we have squeezed some listening test samples into the supplemental and will try to add more later. Thank you for the reminder!
>
> **> Limited fine-tuning evaluation scope**. We agree that the current finetuning analysis is rather narrow. We wanted to respond with preliminary comments as quickly as possible, so we have not yet addressed this at time of writing, but we have committed to adding more experiments with other genres and styles over the course of the rebuttal period. We will update you as we add those to the paper.
>
> **> Baselines.** In Section 5.1, we mention that we do not compare against CA2 due to it being over five times larger by parameter count, like how Qwen 1.5B was not compared against Llama 8B in language modeling, and Qwen 14B was not compared to Llama 70B. When the size difference is that significant, the results are generally uninteresting because the comparison is of apples to oranges. In particular, for DAW integration, we wish for the model to be sufficiently lightweight to run on composers’ existing hardware, which frequently lack dedicated GPUs. CA2, at almost 200M parameters, pushes the bounds of what can run in reasonable time anyway. The model is newer, yes, but it moves away from the particular niche of “edge” device composition that we target, and which CA targeted. [Edit 11/26] Nonetheless, we have added a comparison in Appendix E.5.
>
> We also already provide matched-training-budget (both in time and tokens) LoRA baselines. Could you elaborate on what you would consider “stronger” configurations? We can run them in tandem with the other fine-tuning experiments.
>
> **> Lack of subjective comparison against other models.** We are currently conducting a parallel subjective evaluation of the model against existing baselines, but it will unfortunately not be completed in time for rebuttals. We were unable to do it previously for reasons described in Section 4.3 (Appendix B.4 in the latest revision).
> We would also like to point out that our subjective study is similar in scope to the studies conducted by the similar work MIDI-GPT, Composer’s Assistant, and CA2.
>
> **> Latency gap, arbitrary-mask, and real-time/DAW path.** To address these related points in no particular order: we are currently collaborating with the Calliope team to produce an integration with their DAW, which should address your concern about that, and we should have a prototype by early next year.
>
> We have added a latency comparison on the random infilling objective between CA and our model in the new Appendix E.1; the average latency generally increases from less than 1 second (CA) to just under 5 seconds (ours) on the 8-bar objective and both average times approximately double on 16 bars. We believe that such a slowdown is not significantly important in practice, as it remains a negligible amount of time, given the work produced by the model for its user; see similar waiting times proving a non-issue in Tchmeube et al. [1]. We may move these results to the main text contingent on space restrictions.
>
> **> Multi-track evidence.** It is unfortunately infeasible for us to add any further subjective tests, as that would require us to get ethical approval from our IRB, which cannot happen in time for rebuttals; but we hope the increased fine-tuning experiments on more diverse objectives we plan to perform (i.e. not just melody-only, as you rightfully point out) will speak to “multi-track evidence”. Could you elaborate on what you mean by “attribute controls interacting across tracks”?
>
> **> State-tuning limits.** We flagged the risk as a general statement; the reality of the modern music pretraining landscape is that data is scaling so quickly now (similar to what we saw a few years ago in language modeling) that very little is “extreme”ly out of distribution anymore, now that the largest MIDI datasets have scaled from <200k files (Lakh) to millions and counting (GigaMIDI, Amadeus, etc.). At least, we are not sure what would be extremely OOD. If you had something in mind, please let us know.
>
> [1] Renaud Bougueng Tchemeube et al. Evaluating human-ai interaction via usability, user experience and acceptance measures for MMM-C: a creative AI system for music composition. IJCAI 2023.

---

> ### Author Response · Authors · 2025-11-26
> **Update by Authors**
>
> As we eagerly await your response, we would like to inform you that we have added results on four new finetuning datasets to Appendix D.4: on Pop1K7 (1747 pop pieces), YM2413-MDB (669 videogame soundtracks), the Nottingham dataset (1037 folk songs), and the “waltzes” subset of the Nottingham dataset (35 waltzes). These were also not limited to melody infilling like our POP909 results were. The results are in Tables 10-13 in the appendix. We observe that across various styles and datasets - even in the heavily sample-constrained regime Nottingham “waltzes” subset, training on only 4 songs - state tuning outperforms LoRA finetuning, and we hope these results address your concerns about scope of the finetuning evaluation!
>
> We have also edited our initial response to reflect some other changes made to the submission; in particular, we have added a demo notebook to the supplementary, and have added a comparison against CA2 in Appendix E.5 (although our reasoning as to why we didn’t do this before still stands). We have also expanded our analysis of state tuning with what is, to our knowledge, the first empirical analysis of the mechanics of state tuning in Appendix D.2; we invite you to look at these results as well. We hope these and our previous responses address the vast majority of your concerns!

---

### Author Response · Authors · 2025-11-30
**Final Appeal by Authors**

Given the closure of discussion and the regrettable shifting of the reviewing workload onto the AC, we authors wanted to raise a final public comment to ensure our contributions are not lost. A major virtue of ICLR has always been the ability to raise a paper to acceptance level through hard work during the discussion period, and we would hate for our effort to be wasted due to some bad actors forcing closure of discussions. Therefore this comment summarizes our point of view of the improvements made to the paper, both to facilitate the work of the AC (whom we are sure is swamped) and make a final case for the paper’s acceptance.

We invite the AC to look over the comprehensive responses we authored to each review. We have endeavored to address **every concern** of the reviewers through the addition of **over 12 pages of appendices** to the initial submission (changes marked in magenta) and **updates to the supplementary material**. These include **new baseline comparisons against Composer's Assistant 2 and Polyffusion** (XRdF, mHKN, 1Rax), an **accessible demo** (XRdF, mHKN), **generation examples** (mHKN), further analyses of **generation latency** (XRdF, 4eqD) and **statistical significance** (1Rax), and **general clarifications** (all reviewers). We believe the paper and its contribution have significantly strengthened since the initial poor reviews, largely thanks to the constructive feedback of the reviewers, **all** of which now manifest as additions to the paper.

The most common concern among reviewers, which we strove hardest to resolve, was the *limitation in conceptual novelty of our work*. In this regard, we added **four new datasets to our finetuning evaluation** (all reviewers) that highlight the advantages of our state tuning method in the low-sample regime and suggest it transfers across modalities, and we added an appendix examining the macro- and micro-dynamics of state tuning, which is to our knowledge **the first empirically supported analysis of the dynamics of RNN state tuning** (all reviewers). We invite the AC to explore these in our newly added **Appendices D.4 and D.2**, respectively. We hope these changes, on top of those aforementioned, combine to provide sufficient novelty for ICLR.

We believe these collectively more than suffice to address the concerns raised by each reviewer, and their responses so far seem to broadly indicate agreement. Of the two reviewers who engaged in discussion, **both** indicated willingness to raise their score by at least one or even two levels in their comments, which we are delighted to see! Furthermore, one reviewer who did not participate (4eqD) indicated that "[t]he paper would be **significantly strengthened** if it included experiments to verify" our hypotheses about state tuning dynamics, experiments on which are **newly included in Appendix D.2**.

Finally, we would like to customarily finish the discussion period by thanking all the reviewers for helping improve the paper to this point, as it is undeniably much stronger than the initial submission. We would also like to preemptively thank the AC, regardless of acceptance recommendation, for maintaining the integrity of the review process by taking on such a burden under extraordinary circumstances.

---

### Meta-Review · Area_Chair_b93a · 2026-01-07

**Summary:**

The main concern of the paper is limited conceptual novelty for ICLR, framing it more as a strong application or systems paper than a representation-learning contribution. RWKV itself is an existing architecture, and the paper does not propose a new model and use of music tokenization with bar-based infilling. The proposed controls and training are adaptations of established techniques.

**Reviewer Concerns:**

The rebuttal addressed most technical and empirical concerns by multiple new datasets, including fine tuning beyond melody-only tasks and other system comparisons. Especially interesting was adding a novel empirical analysis of state-tuning dynamics (Appendix D.2), which several reviewers explicitly requested. Subjective evaluation methodology was clarified and some ethics concerns that were raised during the review seem to have been resolved.

**Reviewer Scores:**

The main remaining issue is not correctness or rigor but whether the contribution’s conceptual novelty and general ML insight rise to ICLR standards rather than a domain-focused venue. This concern seem to remain despite improvements and even considering a small bump up in the score, the paper still remains borderline or below threshold. The main theoretical novelty for ICLR would be the state-tuning dynamics in the appendix, but the paper is not written with this analysis as the main focus. The interpretation and empirical evidence that RWKV outperforms LoRA in the very-low-sample regime typical of human composers is intriguing and I would not mind accepting the paper for this reason, but remain conservative in my recommendation.

---

### Decision · Program_Chairs · 2026-01-26

Reject